# Structural basis for the prolonged photocycle of sensory rhodopsin II revealed by serial synchrotron crystallography

Robert Bosman[1,5], Giorgia Ortolani [1,5], Swagatha Ghosh[1], Daniel James [2], Per Norder [1], Greger Hammarin [1], Tinna Björg Úlfarsdóttir[1], Lucija Ostojić[1], Tobias Weinert [2], Florian Dworkowski [3], Takashi Tomizaki [4], Jörg Standfuss [2], Gisela Brändén [1] & Richard Neutze [1] ✉

Microbial rhodopsins form a diverse family of light-sensitive seven-trans-membrane helix retinal proteins that function as active proton or ion pumps, passive light-gated ion channels, and photosensors. To understand how light-sensing in archaea is initiated by sensory rhodopsins, we perform serial synchrotron X-ray crystallography (SSX) studies of light induced conformational changes in sensory rhodopsin II (*Np*SRII) from the archaea *Natronomonas pharaonis*, both collecting time-resolved SSX data and collecting SSX data during continuous illumination. Comparing light-induced electron density changes in *Np*SRII with those reported for bacteriorhodopsin (bR) reveals several common light-induced structural perturbations. Unlike bR, however, helix G of *Np*SRII does not unwind near the conserved lysine residue to which retinal is covalently bound and therefore transient water molecule binding sites do not arise immediately to the cytoplasmic side of retinal. These structural differences prolong the duration of the *Np*SRII photocycle relative to bR, allowing time for the light-initiated sensory signal to be amplified.

Proteins that are closely related structurally often perform very different functions. Understanding how subtle differences in protein structure and dynamics lead to diversity in protein function is one of the goals underpinning the field of structural biology. Rhodopsins are a family of retinal-binding seven-transmembrane (TM) helix proteins that perform diverse biological functions including proton pumping (bacteriorhodopsin, proteorhodopsin, leptosphaeria rhodopsin, xanthorhodopsin and schizorhodopsin), ion pumping (halorhodopsin, the Na$^+$ pump NaR, and the Cl$^-$ pump ClR), membrane depolarization (channel rhodopsin), hyperpolarization (anion channelrhodopsin), and sensory reception (sensory rhodopsin) in bacteria, archaea and algae[1,2]. Visual rhodopsin, another seven TM helix retinal-binding

protein, is a G protein-coupled receptor which serves as the primary light-receptor in animal vision. This evolutionary diversity highlights how proteins utilize similar scaffolds for various functions[3].

Bacteriorhodopsin (bR) from *Halobacterium salinarum* (Fig. 1a) and sensory rhodopsin II from *Natronomonas pharaonis* (*Np*SRII) (Fig. 1b) are intensively studied model systems which provide insight into how light is utilized for energy transduction or phototaxis, respectively. As with all microbial rhodopsins, both contain a buried retinal molecule that is covalently bound through a protonated Schiff base (SB) to a conserved lysine residue of helix G (Lys205 for *Np*SRII and Lys216 for bR)[4–6]. In bR, light activation causes an ultrafast all-*trans* to 13-*cis* retinal isomerization[7,8] which induces a sequence of protein

[1]Department of Chemistry and Molecular Biology, University of Gothenburg, Göteborg, Sweden. [2]Laboratory of Biomolecular Research, Center for Life Sciences, Paul Scherrer Institut, Forschungsstrasse 111, 5232 Villigen PSI, Switzerland. [3]Laboratory of Femtochemistry, Center for Photon Science, Paul Scherrer Institut, Forschungsstrasse 111, 5232 Villigen PSI, Switzerland. [4]Laboratory of Macromolecules and Bioimaging, Center for Photon Science, Paul Scherrer Institut, Forschungsstrasse 111, 5232 Villigen PSI, Switzerland. [5]These authors contributed equally: Robert Bosman, Giorgia Ortolani. ✉e-mail: richard.neutze@gu.se

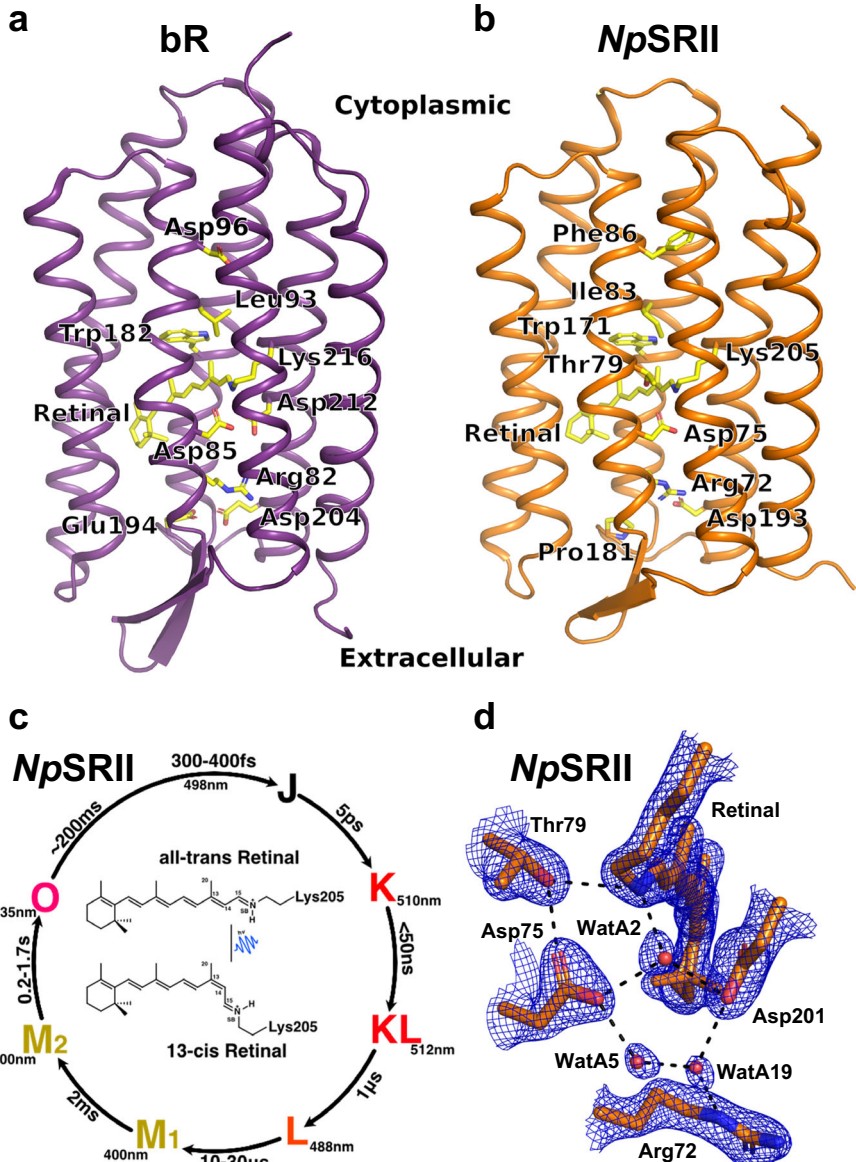

**Fig. 1 | Structures of bR and *Np*SRII and a schematic overview of the *Np*SRII photocycle. a** Structure of bR indicating several of the conserved residues involved in proton pumping. **b** Structure of *Np*SRII with the corresponding residues indicated. **c** Summary of the photocycle of *Np*SRII. Spectral intermediates J to O are labelled, with the wavelength of their absorption maximum and an approximate rise-time indicated (modified from reference[14]). **d** Water mediated hydrogen bond interactions in *Np*SRII connecting the retinal SB to the extracellular side of the protein in the resting conformation. The 2F$^{obs}$-F$^{calc}$ electron density map (blue) is contoured at 1.2 σ, where σ is the root mean square electron density of the map.

conformational changes[9,10] that ultimately return the protein to its starting conformation at the end of the photocycle. Isomerized retinal is deprotonated and Asp85 is protonated during the L-to-M spectral transition[11]. The sequence of structural changes in bR that are required to transport protons against a membrane potential gradient are highly choreographed in space and time[12].

As with bR, SB deprotonation in *Np*SRII occurs during the L-to-M transition[13] (Fig. 1c) with Asp75 on helix C (Fig. 1d, which corresponds to Asp85 of bR) serving as the primary proton acceptor. The M-state of *Np*SRII also forms within a few tens of microseconds, which is similar to bR. By contrast, the M to O transition (~200 ms to 1.7 s[14]) and the decay of the O intermediate back to the resting state (~200 ms) in *Np*SRII (Fig. 1c) is much slower than for bR[13,15], which in the bR photocycle is completed more than an order of magnitude faster[16]. These differences are easily understood from an evolutionary perspective since bR is a proton pump that harnesses light to produce a proton gradient for ATP production, whereas *Np*SRII is a blue light photophobic sensor

that helps protect the cell from harmful near UV-light[17]. A sensory protein requires a prolonged period in the activated state to allow time for the signal to be transduced to downstream receptors and thereby be amplified[18,19]. However, since bR and *Np*SRII are structurally very similar, it is unclear exactly how the second half of the photo-cycle of *Np*SRII is slowed so dramatically[5].

Time-resolved crystallography allows structural changes within proteins to be followed with time[20]. Serial crystallography[21] was first developed at X-ray Free Electron Lasers (XFELs) in order to address the challenge that crystals are rapidly destroyed by a highly brilliant XFEL beam[22]. This motivated the development of new sample delivery methods to continuously feed microcrystals into the X-ray beam path[23] and new approaches to data processing to merge thousands of diffraction patterns containing partial intensities from randomly oriented microcrystals[24]. Serial methods were rapidly adapted for time-resolved diffraction studies[25] and sparked a period of growth in the number of reported time-resolved diffraction studies[26,27]. Serial crystallography

has since been utilized at storage ring facilities[28] with the extension to time-resolved diffraction studies proving an effective way to collect data on slower (milliseconds to seconds) time scales[10].

Here we utilize time-resolved serial synchrotron X-ray crystallography (TR-SSX), and SSX during continuous illumination, to study light-induced structural changes in the archetypal sensory receptor NpSRII. We adapt LCP crystallization procedures for NpSRII[5,6] to facilitate TR-SSX experiments using blue-light illumination and compare structural changes observed after illumination for both NpSRII and bR. These comparisons highlight divergent structural changes underpinning the striking differences in the rate of turnover of their respective photo-cycles. More specifically, our TR-SSX data reveal how the structural rearrangements in helix G near the retinal binding site are considerably smaller for NpSRII than those observed in bR. We suggest that these differences hinder the SB reprotonation step and thereby extend the lifetime of the signalling state of NpSRII. This result provides a structural explanation for how a common protein scaffold can be adapted for varied functionality and highlights the nature of structural insights that may emerge as time-resolved crystallography become increasingly widespread.

## Results and discussion

### Room temperature SSX resting state structure of NpSRII

NpSRII was crystallized using a crystallization precipitant solution modified relative to what has been previously published[5,6]. Notably, we avoided the use of a purple membrane (PM) lipid reconstitution, which allowed large volumes of microcrystals to be produced (a prerequisite for TR-SSX studies) without a corresponding effort to prepare purified PM lipids. Optimization of initial conditions enabled thousands of small NpSRII crystals to be grown in LCP using the well based crystallization approach[29] (Methods). Microcrystals grew as needles with the longest dimension typically up to 60 μm in length, and the smallest dimension <10 μm (Supplementary Fig. 1a, b). However, the longer needles tended to break during handling when LCP microcrystal slurries were transferred into Hamilton syringes for adjusting the sample consistency prior to sample injection, and subsequently loading into the injector reservoir for sample delivery. Thus, prior to injection, microcrystals were typically up to 20 μm in their largest dimension. Serial crystallography data were collected at the Swiss Light Source (SLS) PX1 beamline following the methodology demonstrated using microcrystals of bR[10]. NpSRII microcrystals diffracted to 2.1 Å resolution for the resting (dark) data-collection. When illuminated, these microcrystals diffracted to 2.5 Å resolution (Supplementary Table 1) and this loss in resolution may be due to light-induced movements weakening crystal contacts. The space-group of these microcrystals was C222₁, which had previously been observed for single-crystal structures of NpSRII solved at cryogenic temperature, and both microcrystals and larger crystals have very similar unit cell dimensions: room-temperature microcrystals have a = 89.75 Å, b = 131.7 Å, c = 51.0 Å; cryogenic temperature macrocrystals[5] have a = 89.9 Å, b = 132.6 Å, c = 51.4 Å.

A room temperature SSX structure of NpSRII was solved by molecular replacement using an earlier NpSRII structure[5] for initial phases (PDB entry 1H68). The refined room temperature structure of NpSRII agrees well with the previously published cryo-crystallography structure and differs by only 0.32 Å RMSD on Cα atoms relative to pdb entry 1H68. Whereas high multiplicity serial crystallography data can sometimes reveal unresolved electron density in the loops of integral membrane proteins[30,31], this was not observed for NpSRII since the N-terminal end lacked electron density for the first 20 residues. These findings are in agreement with NMR structures of NpSRII for which the NpSRII tail is intrinsically disordered[32]. Previous cryogenic-temperature structures of NpSRII also highlight that the protein is surrounded by lipid molecules[33]. In our SSX structure we modelled several n-Octyl-β-D-glucopyranoside (BOG) and Monoolein (MPG)

molecules in similar positions to those previously observed (Supplementary Fig. 2).

### Spectral evolution of NpSRII within LCP crystallization conditions

Time-dependent spectra within the UV/Vis domain were measured from NpSRII in LCP crystallization conditions using a modified microspectrophotometer[34]. In this approach, LCP crystallization set-ups after a few days of equilibration (ie. where the sample still appeared homogeneous) were loaded into an X-ray capillary. A blue continuous laser diode (λ = 473 nm) was aligned onto this capillary and a millisecond shutter was used to create pump-laser pulses of 5 ms in duration for time-resolved recordings. Pump-laser pulses of 120 ms in duration were also used to observe a difference spectrum that could be correlated with a photostationary state, since we observed that it takes ~120 ms for the LCP jet to transit through the focused laser spot and into the X-ray beam position. Basis spectra for the photo-intermediates of NpSRII were extracted from the time-dependent sequence of difference absorption spectra (Fig. 2a) and the relative populations of the M and O intermediates were quantified (Fig. 2b) using a linear decomposition script which assumes an exponential decay of the first component into the second component, and of the second component back to the resting state. In this routine, two basis-spectra (for the M and O spectral intermediates, respectively) were optimized as a linear sum of the first two singular value decomposition (svd) components extracted from the set of difference spectra, as has been described in other contexts (section 1.3.3 of supplementary material of reference[35] provides details). The time-dependence of these data show that the relative populations of the M and O intermediates cross-over at Δt = 63 ± 5 ms and that the combined occupancy of M and O has fallen to 50 % of the initial photoexcited population by Δt = 240 ± 10 ms. Each spectroscopy measurement was repeated 45 times on the same sample and there were no signs of light-induced damage until a total laser exposure of several of seconds had accumulated. As such, we are confident that a single 5 ms exposure or during continuous illumination (Fig. 2c) does not generate a detectable fraction of off path reaction species.

Similar measurements were made from slurries of LCP microcrystals (Fig. 2d to f) after the microcrystals had grown (Supplementary Fig. 2a, b), but in this case using ten repeats and with the microcrystals mounted between two glass plates. The quality of these measurements was inferior to that recorded from LCP setups due to the inhomogeneous nature of the sample once microcrystals had formed. Decomposition of these data using the basis-spectra for the M and O intermediates extracted above, showed that the relative populations of the M and O intermediates cross-over at Δt = 820 ± 100 ms and that the combined occupancy of M and O had fallen to 50 % of the initial photoexcited population by Δt = 850 ± 100 ms (Fig. 2e). Thus, once crystals have grown, the SRII photocycle is somewhat slowed, possibly due to the presence of crystal contacts. Irrespectively, these values are within the spread of values experimentally measured for SRII in the scientific literature[13,14,18,36] (Fig. 1c). These occupancies were incorporated into the crystallographic analysis of the TR-SSX data after correcting for the translation of the photoactivated microcrystals in an LCP microjet out of the X-ray beam (Supplementary Table 2).

### Photoactivation of NpSRII microcrystals within an LCP microjet

Light-induced conformational changes in microcrystals of NpSRII were recorded both during continuous illumination (which equates to an approximate exposure of 120 ms due to the time required for the LCP jet to transit into the centre of the laser focus) and after microcrystals were exposed to blue-light flashes 5 ms in duration at a repetition rate of 4 Hz. An LCP flow-rate of 0.2 μl/min and using a 75 μm diameter LCP microjet, allowed the sample to be translated ~190 μm between each 5 ms flash (the average downwards velocity is calculated to be 760 μm/

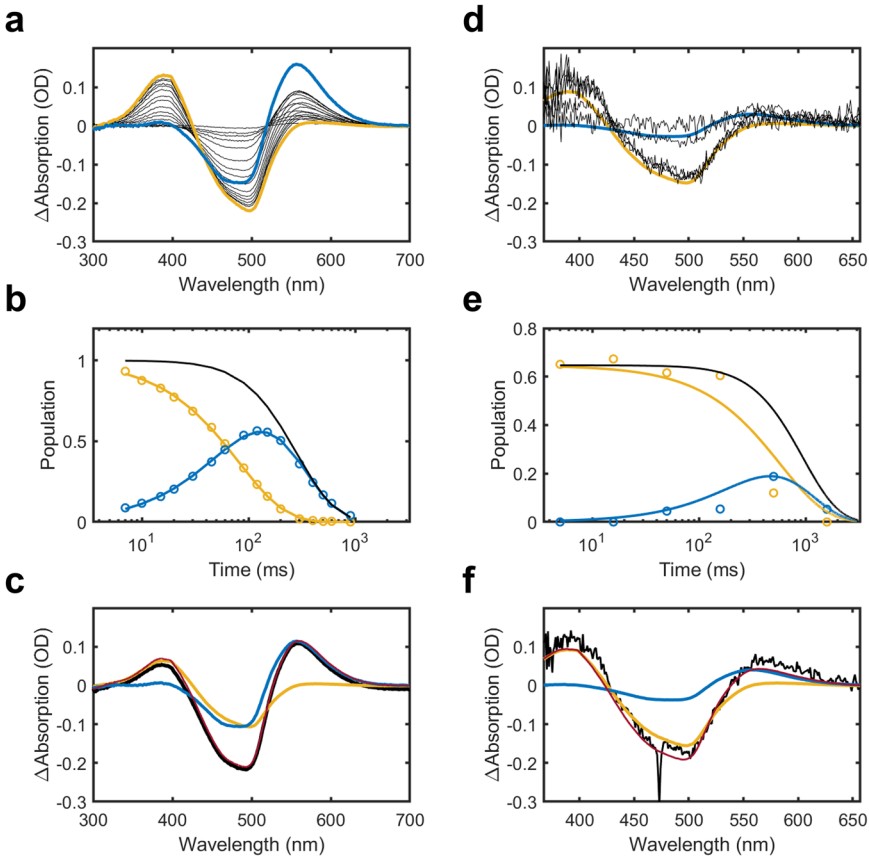

**Fig. 2 | Time-resolved UV/vis spectroscopy measurements of changes in absorption of *Np*SRII. a** Sequence of difference spectra (black lines) recorded from *Np*SRII samples incorporated into LCP crystallization setups. The bold mustard line represents the basis spectrum extracted from these data characteristic of the M-state whereas the bold blue line represents that associated with the O-state. **b** Time-dependence of the components associated with the M-state (mustard line), the O-state (blue line) and the total population of all photoactivated species (black line). These data are normalized to a total photoexcited population of 1 when projected back to $\Delta t = 0$ ms. **c** Difference spectrum associated with the photostationary state after 120 ms of illumination (black line) and the breakdown of this difference spectrum into M (mustard) and O (blue) states. **d** Sequence of difference

spectra (black lines) recorded from LCP microcrystals of *Np*SRII plotted on the same scale as in panel (**a**). Basis spectra (mustard, M-state; blue, O-state) are the same as those derived in (**a**). **e** Time-dependence of the components associated with the M-state (mustard line), the O-state (blue line) and the total population of all photoactivated species (black line) in microcrystals. **f** Difference spectrum associated with the photostationary state in microcrystals after 120 ms of illumination (black line) and the breakdown of this difference spectrum into M (mustard) and O (blue) states. Quantified occupancy values used for structural refinement extracted from this analysis are provided in Supplementary Table 2. Source data are provided as a Source Data file.

sec) and the same region of the LCP microjet was never exposed to blue light more than once. A 5 ms exposure using a continuous 2.6 mW laser focused into a 75 µm FWHM (full width half maximum) diameter spot yields the dimensionless product $F \cdot \sigma / h\nu \sim 49$ when incident upon the microjet surface and assuming no experimental losses due to scatter, and this value is ~37 when averaged throughout the volume of a representative microcrystal, where $F$ is the laser pulse fluence averaged across the FWHM focal spot, $\sigma$ is the absorption cross section of the resting state of *Np*SRII, and $h\nu$ is the product of Planck's constant with the frequency of the light. Under these conditions, we observe the evolution of structural changes from a photo-stationary equilibrium rather than structural changes induced by a single-photon absorption event. Although microcrystals will be heated to some extent by the energy of the absorbed light, during continuous illumination these crystals continued to diffract. Some heating induced disordering might be implied by the observation that the average B-factor is 54.0 Å² for the dark data-set and 60.0 Å² during continuous illumination (Supplementary Table 1). However, the average B-factor for the time-resolved data sets from 0 –150 ms vary from 45.0 Å² – 47.0 Å² (Supplementary Table 1). Since these values are lower than the dark data-set, sample to sample variation in crystals may also explain this observation.

Following each laser exposure, 24 sequential bins of 10 ms in duration were recorded using an Eiger X-ray detector[37] with an experimental setup similar to that previously reported[10] (Methods, Supplementary Fig. 1c, d). X-ray diffraction data were then combined into sets corresponding to eight 30 ms time-windows, of which five (from 0 to 150 ms) were judged to have sufficiently high population of the photoactive state for further structural analysis (Supplementary Table 1). Long distance overviews of the difference Fourier electron density changes induced by light in *Np*SRII are provided for two representative time-delays $\Delta t = 0$ –30 ms (Fig. 3a) and $\Delta t = 120$ –150 ms (Fig. 3b), and during continuous illumination (Fig. 3c). This representation makes apparent that all *Np*SRII difference Fourier electron density maps are very similar and there are recurring themes when compared with light-induced conformational changes in bR (Fig. 3d)[10]. By representing difference Fourier electron density throughout a three-dimensional structure as a one-dimensional mapping onto the protein sequence[38], (Fig. 4a) we observed that structural changes from 0 ms to 120 ms are highly-correlated and have internal Pearson correlation coefficients from 51% –71% and that this correlation drops to as low as 13% as the signal-to-noise falls on longer time-scale (Supplementary Fig. 3). The time-scale of this loss of correlation implies that photoactivated *Np*SRII microcrystals were swept out of the X-ray beam

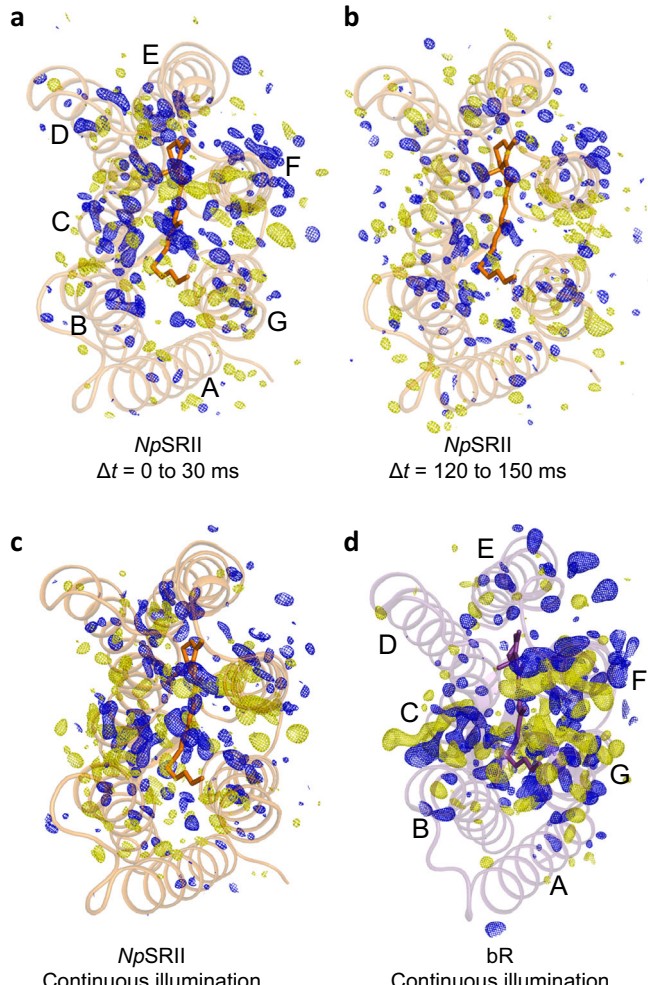

**Fig. 3 | Overviews of the $F_o$(light)−$F_o$(dark) isomorphous difference Fourier electron density maps for $Np$SRII and bR. a** Difference electron density map calculated from TR-SSX data collected for $\Delta t = 0$–30 ms. **b** Difference electron density map calculated from TR-SSX data collected for $\Delta t = 120$ –150 ms. **c** Difference electron density map calculated from SSX data collected during continuous illumination of $Np$SRII microcrystals. **d** Difference electron density map calculated from SSX data collected during continuous illumination of bR microcrystals[10]. All $F_o$(light) − $F_o$(dark) difference Fourier electron density maps (blue, positive difference electron density; yellow, negative difference electron density) are contoured a ± 3.0 σ, where σ is the root mean square electron density of the difference Fourier map.

~120 ms after photoactivation. Consequently, the portion of the LCP microjet above the X-ray beam that is photoactivated by the laser flash is ~90 μm high (120 ms × 760 μm/sec), which is significantly larger than the nominal focal radius of 38 μm (FWHM/2) should the laser beam and X-ray beam be perfectly overlapped. As such, there must be sufficient light to photoactivate microcrystals across a larger radius than defined by the FWHM and internal scattering within the microjet may have broaden the focal spot to some extent.

**Time-evolution of electron density changes in NpSRII**
From this one-dimensional mapping, it is apparent that all major structural changes are present for the time-points 0 ms ≤ $\Delta t$ ≤ 120 ms (Fig. 4a). Partial occupancy structural refinement succeeds in capturing the nature of these motions (Fig. 4b). For Np$SRII$, the strongest electron density changes are associated with helices C, F, and weaker difference density features are associated with the extracellular loop linking helices D and E. From the time-evolution of these changes, we

observe a steady decay of difference electron density features associated with both helices C and F from 0–120 ms (Fig. 4c). By contrast, difference density features associated with helix G are weaker, yet are relatively constant until $\Delta t$ ≥ 120 ms, which is the time-scale when the photoactivated sample is swept from the X-ray beam position. Observations from the photostationary state (Fig. 4a, c) show that the difference electron density features associated with helix C are more pronounced that those associated with helix F during continuous illumination (Fig. 3a–c). From a functional perspective, the thermally driven re-isomerization of retinal from 13-$cis$ to all-$trans$ configuration, which occurs after the Schiff-base reprotonation associated the M to O transition but before the structural relaxation[39] of helix F, would allow WatA2, WatA5 and WatA19 (Fig. 1d) to reorder and thereby return helix C to its resting conformation. During continuous illumination, retinal will be undergoing both forward and backward photoisomerization. This would cause this extracellular water structure to remain disordered and thereby exaggerate the extent to which the extracellular portions of helix C are able to flex towards helix G, hence the relatively strong difference electron density associated with helix C during continuous illumination.

**Light-induced rearrangements of extracellular water networks and helix C**
Examining the difference Fourier electron density maps for $Np$SRII calculated from data collected during continuous illumination minus data collected without illumination (Figs. 3c, 5b, 6b) reveals both similarities and differences for light-induced changes in electron density in $Np$SRII when compared to data collected from bR during continuous illumination[9,10] (Figs. 3d, 5a, 6a). Time-resolved serial femtosecond X-ray crystallography (TR-SFX) studies show that Wat402 in bR is slightly displaced by retinal isomerization on a sub-picosecond time-scale[7,8] and becomes disordered on a nanosecond time-scale[9]. Wat402 serves as a structural keystone in bR since its disordering initiates a cascade of structural changes[40] that evolve with time throughout the protein. Evidence for this structural cascade is captured during continuous illumination of bR crystals[10] as strong negative electron density on Wat402 (-6.8 σ, where σ is the root mean square electron density of the unit cell), Wat401 (-6.1 σ) and Wat400 (-3.6 σ) and a positive feature between these water molecules (4.6 σ, Fig. 5a) corresponding to the ordering of Wat451 (Fig. 5d).

As with bR, a negative difference electron density feature can be seen in the difference Fourier electron density map calculated from continuous illumination of $Np$SRII microcrystals which indicates that the corresponding keystone water molecule (WatA2, -4.1 σ) is disordered (Fig. 5b, Supplementary Fig. 4). Negative electron density features are also observed on neighbouring water molecules (WatA5 and WatA19, -5.2 σ and -3.2 σ respectively) and a positive electron density feature arises indicating the transient ordering of a new water molecule (WatB5, 5.1 σ) in $Np$SRII. These same features are visible for $Np$SRII in the time-resolved sequence of difference Fourier electron density maps for the time-delays 0 ms to 150 ms, but with the relative intensities varying due to experimental noise associated with these data (Supplementary Fig. 5). For example, the time-delay $\Delta t = 60$−90 ms shows negative difference electron density features on WatA2 (-3.6 σ) and WatA5 (-3.7 σ) and a positive density feature arises (Fig. 5c) which was modelled as WatB5 (3.1 σ, Fig. 5e). Moreover, both bR and $Np$SRII show that these water rearrangements disrupt the conserved arginine residue (Arg82 in bR; Arg72 in $Np$SRII) indicating that this side-chain reorients towards the extracellular space during photocycle-turnover. This, in turn, displaces a chloride ion that interacts with Arg72 and Tyr 73 in the resting SRII conformation[5]. These specific disruptions of the extracellular H-bond network in $Np$SRII at room-temperature were also observed in low-temperature intermediate trapping studies[41–44].

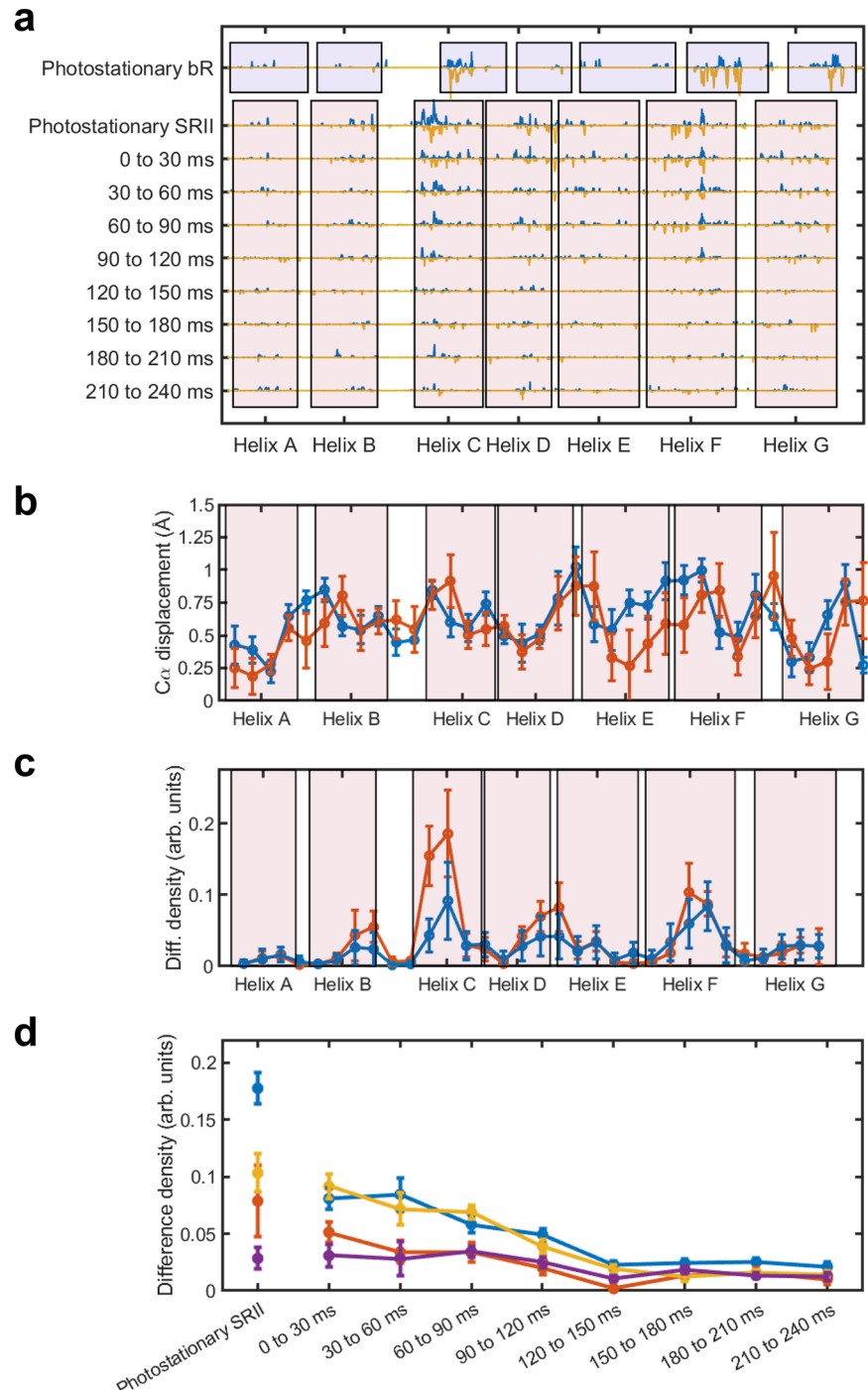

**Fig. 4 | Time-dependence of difference electron density features in *NpSRII*.**
**a** Difference electron density (blue positive density, mustard negative density, arbitrary units) is represented as a function of sequence according to the procedure described by Wickstrand et al.[38] Similar difference electron density features are observed for bR and *NpSRII* during continuous illumination, and within the time-resolved data following a 5 ms laser-flash. Transmembrane helices of bR are indicated in transparent purple and those of SRII in transparent pink. Helix boundaries for bR are helix A, residues 6 –32; B, 37 –58; C, 80 –100; D, 105 –127; E, 131 –160; F, 165 –191; G, 201 –224. Helix boundaries for SRII are helix A, residues 3 –26; B, 33 –56; C, 70 –92; D, 94 –118; E, 122 –150; F, 153 –181; G, 189 –219. **b** Root mean square deviations (rmsd) of Cα atomic displacements after partial occupancy refinement are shown averaged over the time-delays $\Delta t = 0$ ms to 90 ms (blue line) and for the photostationary state (red line). Coordinate error bars are estimated from procedures developed in reference 64: 100 sets of refined coordinates are generated from 100 sets of bootstrap resampled data, and the standard-deviations (error bar

estimates) of the mean separations between coordinates from light and dark data are calculated. **c** Quantification of difference electron density features averaged over the time-delays $\Delta t = 0$ ms to 90 ms (blue line) and for the photostationary state (red line). Error bars associated with difference electron density observations are estimated from procedures developed in references 64 and 38: difference electron density maps are calculated from 100 sets of bootstrap resampled light and dark data, and the standard deviations (error bar estimates) of difference electron density features are quantify. **d** Time-dependence of the mean amplitude of difference electron density changes associated with the extracellular portions of helix C (Pro71 to Pro81, blue data), the extracellular portions of helices D and E (Val118 to Tyr124, red data), the cytoplasmic portions of helix F and the F/G loop (Gly143 to Pro171, mustard data), and the mid-portions of helix G (Ile197 to Leu213, purple data). Error bars associated with difference electron density observations are estimated using the same procedures as in panel c. Source data are provided as a Source Data file.

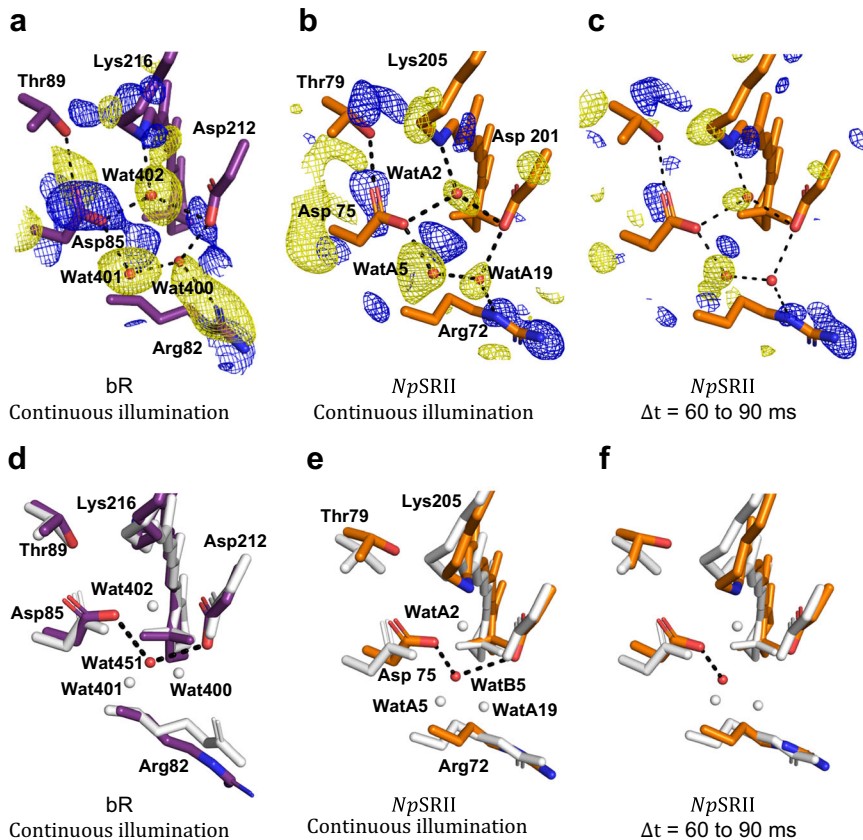

**Fig. 5 | Light-induced rearrangements on the extracellular sides of the active-sites of bR and *Np*SRII. a** Difference electron density observed for bR during continuous illumination indicates the light-induced disordering of active-site water molecules and movements of the side-chain of Lys216, Arg82, Asp85 and Thr89. **b** Difference electron density observed for *Np*SRII during continuous illumination indicates the light-induced disordering of active-site water molecules and movements of the side-chain of Lys205, Arg72, Asp75 and Thr79. **c** Difference electron density observed for *Np*SRII for $\Delta t = 60–90$ ms indicates similar changes in electron density as observed in panel (**b**). Whereas paired positive and negative difference electron density features indicate similar structural rearrangements for these conserved water molecules and residues, there are notable differences associated with the Asp75 and Thr79 side-chains in *Np*SRII relative to Asp85 and Thr89 in bR. All $F_o(\text{light}) − F_o(\text{dark})$ isomorphous difference Fourier electron density maps are contoured at $\pm 3.0\,\sigma$, blue, positive difference electron density; yellow, negative difference electron density. **d** Light-induced conformational changes in bR resulting from structural refinement against dark (white) and continuous illumination (purple) data. **e** Light-induced conformational changes in *Np*SRII resulting from structural refinement against dark (white) and continuous illumination (orange) data. **f** Light-induced conformational changes in *Np*SRII resulting from structural refinement against dark (white) and data recorded for $\Delta t = 60$ to 90 ms (orange).

It has long been known that the rearrangement of these water molecules along the extracellular proton-translocation channel are a prerequisite for helix C to move towards helix G in bR[9,45,46] and similar motions of helix C towards helix G are apparent in the difference electron density recorded for *Np*SRII (Figs. 3a–c, 4). TR-SFX studies of bR show that this movement arises on the same-time scale as the SB deprotonation[9] and consequently the flexing of transmembrane helix C was argued to be the rate-limiting step that governs the primary proton transfer event[9,45,46]. As such, we conclude that the inward flexing of helix C is also an essential part of the structural mechanism governing proton-exchanges in *Np*SRII[47]. Despite these similarities, it is noteworthy that the difference electron density associated with Asp85 of bR and Asp75 of *Np*SRII are distinct. For bR, there is a strong positive difference density feature that indicates a rotation of the side chain of Asp85 away from Thr89 and this motion breaks their mutual hydrogen bond (Fig. 5a, d). Moreover, in TR-SFX studies of bR, transient negative difference electron density was observed between Asp85 and Thr89, which illustrates how the Asp85-Thr89 H-bond is broken, arose as the SB became deprotonated[9]. By breaking this connection once Asp85 was protonated, the pathway for reverse proton-exchange from Asp85 to the SB via Thr89 is disrupted and hence this futile back-reaction is prevented. Difference electron density features associated with the Asp75 and Thr79 side-chains of *Np*SRII are consistent with these residues moving in unison (Fig. 5b, c), which is captured to some extent by structural refinement but this also involves to a rotation of the Asp75 side-chain as WatB5 orders and the Asp75-Thr79 H-bond is broken (Fig. 5e, f). Proton-transfer steps have been identified in the photocycle of *Np*SRII and this light receptor has been shown to translocate protons but cannot pump against a strong membrane potential[48,49]. Other studies have suggested that proton uptake and release steps may also occur from the extracellular side of the protein and therefore there is no net transfer of protons[47]. Differences in the evolution of structural changes associated with the conserved aspartate proton acceptor, and in particular the nature of correlated motions of Asp75 and Thr79 (Asp 85 and Thr89 in bR), may help understand why bR is an effective proton pump whereas *Np*SRII is not.

## Light-induced movements of the extracellular regions of helices D and E

Inspection of the difference electron density maps (Fig. 3) and their representation in 1D (Fig. 4) shows that *Np*SRII has relatively weak but recurring electron density changes associated with the extracellular regions of helix D and E and these features are more extensive than what is observed in bR. These motions may be due the packing of helix

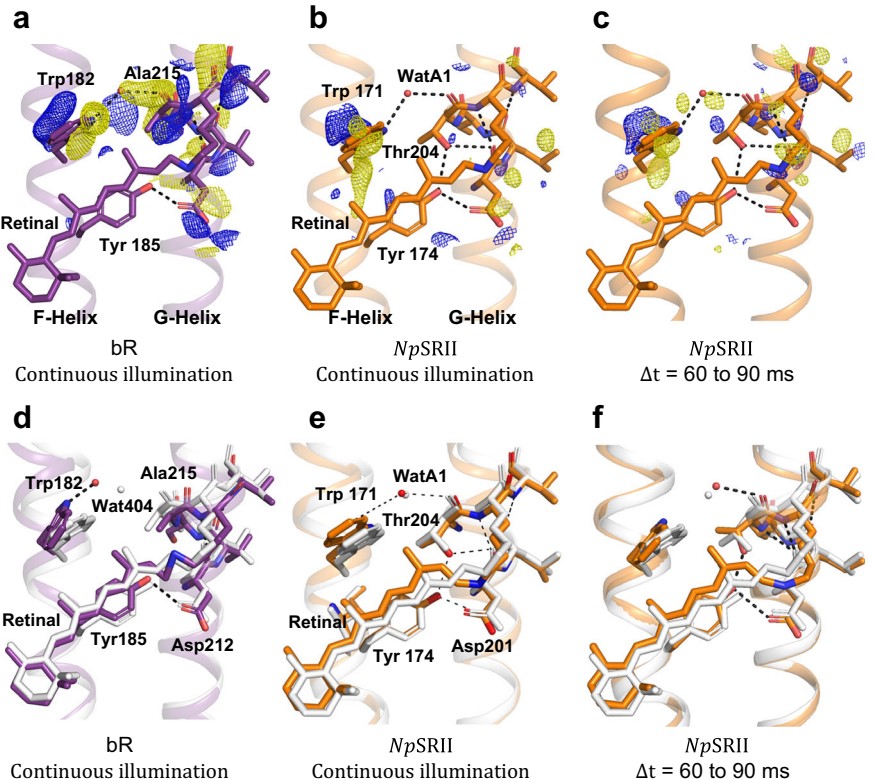

**Fig. 6 | Light-induced rearrangements in helices F and G on the cytoplasmic sides of of bR and *Np*SRII. a** Paired positive and negative difference electron density features observed for bR during continuous illumination indicate a light-induced movement of Trp182 and Ala215. **b** Paired positive and negative difference electron density features observed for *Np*SRII during continuous illumination indicate the light-induced movement of Trp171, but there is no corresponding motion associated with Thr204. **c** Paired positive and negative difference electron density features observed for *Np*SRII for $\Delta t = 60$ to 90 ms indicate similar changes in electron density as observed in panel (**b**). Both $F_o$(light) − $F_o$(dark) difference

Fourier electron density maps are contoured at ±3.0 σ, blue, positive difference electron density; yellow, negative difference electron density. **d** Light-induced conformational changes in bR resulting from structural refinement against dark (white) and continuous illumination (purple) data. **e** Light-induced conformational changes in *Np*SRII resulting from structural refinement against dark (white) and continuous illumination (orange) data. **f** Light-induced conformational changes in *Np*SRII resulting from structural refinement against dark (white) and data recorded for $\Delta t = 60$–90 ms (orange). These representations indicate that movements in helix G following retinal isomerization are much reduced in *Np*SRII relative to bR.

D against helix C, and consequently these regions move as the extracellular domain of helix C moves. In this context it is noteworthy that helix E is slightly displaced in *Np*SRII relative to its position in the bR resting state structure[5,6]. Photosensory proteins must control the chromophore's photo-absorption maximum in order to ensure the appropriate response to light of different wavelengths and chimeric studies have indicated that helices D and E influence the spectral tuning in *Np*SRII[50]. For these reasons, tighter interactions between helices C, D and E are required for spectral tuning and these may induce more correlated motions of these helices during the *Np*SRII photocycle.

### Light-induced movements of helix F
Both bR and *Np*SRII show paired positive and negative difference electron density features on helix F during continuous illumination or after a 5 ms laser pulse (Figs. 3, 4, 6, Supplementary Fig. 5). These features indicate an outward movement of the cytoplasmic portion of helix F for both proteins. The corresponding difference electron density features are considerably stronger in bR than in *Np*SRII and this is borne out by structural refinement (Supplementary Fig. 6), for which the magnitude of Cα-displacements for helix F are 0.71 Å in *Np*SRII (averaged from Pro144 to Pro175) and 3.78 Å in bR[10] (averaged from Gly155 to Pro186; pdb entries 6RQP, dark conformation; 6RPH, open conformation). Retinal geometry governs these movements since the 13-*cis* configuration causes the retinal's C20 methyl group to come into

steric conflict with a conserved tryptophan residue (Trp182 in bR and Trp171 in *Np*SRII), displacing it towards the cytoplasm and thereby driving an outward movement of transmembrane helix F, as has been observed by TR-SSX[10], TR-SFX[9], time-resolved X-ray solution scattering[35] and spin labelled EPR spectroscopy[51]. Whether or not the difference in the amplitude of these motions observed for bR and *Np*SRII is physiological, or due to differences in crystal packing, cannot be determined from the TR-SSX data alone.

### Light-induced rearrangements within helix G
A striking difference between the bR and *Np*SRII photo-activated states is visible in helix G. In both *Np*SRII and bR resting conformations, a conserved tryptophan side-chain coordinates a water molecule positioned between helices F and G (Fig. 6). This water molecule interacts with Nε of Trp171 (SRII) and the backbone carbonyl oxygen of Thr204 in *Np*SRII, and Nε of Trp182 (bR) and the backbone carbonyl oxygen of Ala215 in bR. In bR, strong paired positive and negative difference electron density features (Fig. 6a) stretch from Trp182 (paired positive and negative density features on this sidechain of 5.8 σ and -5.2 σ), through Wat404 to Ala215 in a correlated motion that involves the partial unwinding of helix G (Fig. 6d). These motions arise from a steric conflict between the C20 methyl and Trp182 as retinal isomerizes, and the physical pulling of Lys216 on helix G. Thus, helices F and G move in opposite directions and the H-bond water bridge between these two helices breaks, which partially opens up the cytoplasmic portion of the

proton transport channel and thereby aids reprotonation of the SB from Asp96. Moreover, the unwinding of helix G in bR frees the carbonyl oxygen of Lys216 (paired positive and negative density features on this carbonyl oxygen of 7.6 σ and -6.0 σ) from its H-bond interaction to the amide nitrogen of Gly220 and provides a site for water a molecule (Wat453) to transiently bind[10]. Along with Wat404 and Wat454 (numbered according to pdb entry 6RPH), this transiently ordered water molecule participates in a water-mediated H-bond pathway from the SB to the proton uptake group (Asp96) on the cytoplasmic side of the proton transport channel[9,10,52], further aiding SB reprotonation in bR.

By contrast, difference electron density recovered from photo-activated microcrystals of *Np*SRII show weak negative density between Trp171 and Thr204 that does not overlap with near WatA1, both during continuous illumination (Fig. 6b) and for the sequence of time-points following the 5 ms laser pulse (Fig. 6c, Supplementary Fig. 5). Moreover, there are no paired difference electron density features to indicate that helix G moves away from helix F (Fig. 6 b, c), and the difference features associated with the carbonyl oxygen of Lys205 during continuous illumination are weak (3.1 σ and -3.4 σ respectively) when compared with features on Trp171 (4.4 σ and -4.5 σ respectively). Partial occupancy refinement confirms that WatA1 of *Np*SRII maintains H-bonds to both Trp171 and Thr204 during continuous illumination (Fig. 6e). Structural refinement suggests that the Trp171-WatA1 H-bond is lost for $\Delta t = 60$ to 90 ms (Fig. 6f) but this may be due to the limitations of partial occupancy refinement when trying to place two overlapping water-molecules with complementary occupancy at 2.4 Å resolution. Furthermore, it is observed that helix G of *Np*SRII does not unwind, possibly due to the H-bond pattern of the Thr204 side-chain providing additional stability for helix G in this region, with Oγ of Thr204 forming hydrogen bonds to the side chain oxygen of Tyr174 of helix F and to the carbonyl oxygen atoms of Leu200 and Asp201 of helix G (Fig. 6). Fourier Transform Infrared (FTIR) spectroscopy indicates that the Tyr174-Thr204 hydrogen bond is stretched early in the *Np*SRII photocycle, but is restored in the M-state[53]. Consequently, the carbonyl oxygen of Thr204 does not create a transient water-binding motif to facilitate proton transfer from the cytoplasm, and the interface between helices F and G cannot open up to the same extent as observed for bR. Consistent with these observations, EPR and FTIR data also indicate that helix F of *Np*SRII moves whereas helix G does not[18,39]. SB reprotonation from the cytoplasm in *Np*SRII is therefore delayed relative to bR, and these observations can explain the very extended lifetime of the M-intermediate in the photoreceptor. Furthermore, the Thr204Ala mutation in combination with other site-specific mutations can accelerate the *Np*SRII photocycle by more than two orders of magnitude[54] and Thr204 has been shown to be essential for light signalling in the *Np*SRII:HtrII complex[55]. These observations also assign a functional significance to the π-bulge observed in helix G of bR in this region[56], for which Ala215 does not partake in the usual n + 4 H-bond pattern of an α-helix. This π-bulge weakens this transmembrane helix in order to assist its transient unwinding. Although a similar π-bulge is observed for Thr204 in the resting conformation of *Np*SRII, the additional H-bond interactions of the tyrosine side-chain stabilize helix G in this region.

### Functional implications

Comparative time-resolved diffraction studies yield a structural perspective on how subtle differences in protein structure influence protein function. Since many protein catalysed reactions occur on time-scales from microseconds to seconds, TR-SSX studies at synchrotron radiation facilities should allow time-resolved diffraction to be applied to a much broader set of biochemical questions than what was traditionally accessible using time-resolved Laue diffraction[20,27]. Serial crystallography data collected during continuous illumination of *Np*SRII and bR highlight the potential of this approach, since these structural observations emphasize how the relative flexibility of helix G controls the duration of the photocycle. Efficient proton pumping requires rapid turnover whereas light-signalling requires an extended lifetime of the signalling state. As such, the relative flexibility near the centre of helix G strongly influences these functionalities and highlights the importance of Thr204 in extending the duration of signalling states of *Np*SRII. These findings illustrate how light-induced electron density changes observed by TR-SSX reveal both common and diverging structural themes governing the dynamics of these two retinal proteins with distinct functions. Such advances may usher in a new perspective for time-resolved diffraction, which has the potential to become a widely used biophysical method for describing a broad sphere of protein catalysed reactions[27].

## Methods

### Protein production

*Np*SRII was purified as previously reported[42]. In brief, His-tagged *Np*SRII was expressed in BL21star *E-coli* cells in pET27a expression vectors for 2.5 hr after induction with 10 mM IPTG. Cells were then pelleted, resuspended in T-buffer (50 mM Tris, 5 mM MgCl₂, 2 mM EDTA, lysozyme, EDTA-free Protease Inhibitor Cocktail, Roche, pH8) and cells lysed by sonication. Lysate was bound to a Ni-NTA column. For every gram of cells, we used 1/2 ml of resin, which establishes the column volume, and washed twice (with 50 mM MES, 300 mM NaCl, 1% β-OG, pH 7.5 and 40 mM imidazole on the second step). A final elution step used 50 mM Tris, 300 mM NaCl, 160 mM imidazole. *Np*SRII containing fractions were determined as those having a specific absorption peak at 498 nm. These were pooled, concentrated and run on a size exclusion column, HiLoad 16/600 Superdex 200 pg column. High purity was determined by a 280:498 ratio of 1:1.2 and these fractions were collected for crystallization.

### Crystallization

SRII was concentrated to 1.6 mM and reconstituted into LCP with monoolein (9.9 MAG) using coupled syringes. The precipitant solution used was CaCl₂ 150 mM, Glycine 100 mM, 38% ($v/v$) PEG 400, pH 7.5. Crystals were grown in ~200 μl LCP in glass deep-well plates[29] with 10 μl protein containing LCP and 400 μl precipitant solution over ~1–2 months. These setups yielded crystals 40-60 μm in their largest dimension, which were suitable for SSX. The crystal-containing LCP was harvested and packed into a 1 ml Hamilton syringe for transport.

### Spectroscopic characterization

Time-resolved spectra from *Np*SRII were recorded using a modified single crystal microspectrophotometer[34] from crystallization setups held in a 300 μM glass capillary. A 20 mW continuous blue-light laser ($\lambda = 473$ nm) was focused to a FWHM spot ~200 μm × 400 μm and a millisecond shutter was used to control the pulse duration. The time-delay between the millisecond exposure and when a microsecond flash-lamp was triggered was controlled using a pulse generator. Each pump-probe measurement was repeated 45 times when recording from LCP setups, and repeated 10 times when recording from LCP microcrystals. Outliers were rejected, and the remaining data were averaged to improve signal to noise (Fig. 2). Indications of light induced-damage began to be observed as a stable species absorbing near 400 nm after an accumulated laser exposure time in excess of 10 s. Since this laser intensity was similar (approximately a factor of 2 lower intensity) to that used during time-resolved serial crystallography measurements, and since each microcrystal was exposed at most to a single 5 ms (time-resolved) or ~125 ms (continuous illumination) laser pulse, the population of light damaged *Np*SRII molecules within microcrystal is considered far below levels detectable using X-ray crystallography.

## Data collection

We used an experimental setup that was similar to that used for earlier studies of bR[10]. The sample's viscosity was optimized to achieve a stable LCP extrusion by the addition of 20%–25% monoolein to the harvested microcrystal batches. Microcrystals were injected at ~0.2 µl/min from a 20 µl reservoir using a 75 µm nozzle by an LCP extruder designed by Arizona State University[57]. X-ray diffraction data were collected on an Eiger 16 M hybrid pixel detector. Samples were photo-excited using a 488 nm Class3R laser diode with a measured power of 2.6 mW at the sample position, which was focused into a FWHM spot of 75 µm × 75 µm. The X-ray beam position was located and aligned using a YAG screen (which is fluorescent when exposed to X-rays) and a high-magnification camera. The focal spot size of the laser diode was determined offline using a knife-edge scan. The laser diode was aligned on the extruding jet using an attenuated beam that allowed centring of the laser spot vertically by centring the maximal intensity on the X-ray beam. Horizontal alignment was aided by laser scattering from the LCP jet, being aligned to produce a symmetric speckle pattern behind the jet. The X-ray energy was 12.4 keV focused to a spot size 20 µm horizontal and 5 µm vertical. For photo-stationary measurements, the laser illumination at the X-ray position was continuous and 30 min of 'light' data were collection followed by 30 min of 'dark' data collection with a 20 ms frame-rate. For time-resolved studies, the laser diode and detector where trigger via a TTL pulse giving a 5 ms laser pulse. Twenty-four frames of 10 ms in duration were collected sequentially and the cycle was repeated. Under these conditions, applying an extinction coefficient σ for $Np$SRII of 48 000 cm$^{-1}$Mol$^1$ suggests that the dimensionless product[27] σ·F/hν ~ 49 when incident upon the microjet surface and ~37 when averaged over a 20 um long microcrystal (O.D. = 0.26) since the laser fluence $F = 147$ mJ/cm$^2$ when averaged across the FWHM for each 5 ms exposure, where h is Planck's constant and $v = 488$ nm. $Np$SRII should thus be considered as having evolved from a photo-stationary state for the time-resolved studies, or be captured in a photo-stationary state during continuous illumination.

## Data processing

Diffraction peaks were identified using indexamajig from CrystFEL0.8.0[24] using peakfinder8 and indexed using the xgandalf algorithm[58]. X-ray diffraction data contained two crystal forms, with the majority of images diffracting to only very low resolution. When the unit cell axes a ≈ 89.8 Å, b ≈ 131.7 Å, c ≈ 51.0 Å were imposed, this sub-set of data diffracted to much higher resolution, and this crystal form was selected throughout (Supplementary Table 1). Integrated data were merged using the programme partialator using custom-split-list to improve scaling. Time-resolved diffraction data were divided into 24 bins of 10 ms each, which were in turn pooled into the following time-bins: 0–30 ms; 30–60 ms; 60–90 ms; 90 - 120 ms; 120–150 ms; 150–180 ms; 180–210 ms; and 210–240 ms, with each bin containing ~7500 indexed images. Datasets were scaled and measured X-ray diffraction intensities were converted to structure factor amplitudes with appropriate errors using CCP4 TRUNCATE[59,60]. Molecular replacement using pdb entry 1H68[5] for phases was used to recover the room temperature SSX structure. The structural model was improved using multiple rounds of refinement in Phenix and manual manipulations in COOT[61]. Resting state phases from the refined model were used for difference Fourier electron density map calculations (FT[($F_o$(light) − $F_o$(dark)) × exp($i\Phi_{dark}$)]), FT = Fourier Transform and $\Phi_{dark}$ represent experimental phases calculated from the dark-state model, using the Phenix isomorphous $F_o$(light) − $F_o$(dark) calculator[62]. Difference Fourier electron density maps were interpreted directly, and the sensitivity of features on WatA2 to the resolution cut-off was examined (Supplementary Fig. 4). Observed light induced changes in electron density were analysed using a correlation script[38] (Fig. 4, Supplementary Fig. 3). Light illuminated structures were refined using partial occupancy refinement with an occupancy specified in Supplementary Table 2 and the complementary resting conformation restrained about the dark conformation. During refinement, the population of the M-state was given a 13-$cis$ retinal, whereas the resting and O-state populations were refined with an all-$trans$ retinal (Supplementary Table 2). Intermediate state structures were refined with PHENIX[62]. Atomic coordinates (x, y, z) and individual B-factors were refined for all atoms, and their occupancies were fixed based upon information from spectral characterization (Supplementary Table 2). Water molecules that were associated with negative peaks in isomorphous difference Fourier electron density maps were removed from the B conformer of the protein. Refinement statistics can be found in Supplementary Table 1.

## Reporting summary

Further information on research design is available in the Nature Portfolio Reporting Summary linked to this article.

## Data availability

Coordinates and structure factors for the SSX $Np$SRII dark and continuously illuminated structures have been deposited in the RCSB Protein Data Bank under accession codes 9H20(dark conformation) and 9H1X(continuous illumination), 8PWP (0–30 ms), 8PWJ (30–60 ms), 8PWI (60–90 ms), 8PWG (90–120 ms), 8PWQ (120–150 ms), and 9H1Wfor the grouped dark time-points (150–240 ms). Source data are provided with this paper.

## Code availability

In-house code written in Matlab used to generate Fig. 2 has been placed on Github (https://github.com/Neutze-lab/TR_SSX_SRII) and at Zenodo repository, entry 14916023[63]: Resampling scripts for serial crystallography were described in reference[64] and have been placed on Github (https://github.com/Neutze-lab/resampling) and at Zenodo repository, entry 14921799[65].

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

## Acknowledgements
We acknowledge the Paul Scherrer Institut, Villigen, Switzerland for provision of synchrotron radiation beamtime at beamline PX1 of the SLS. We thank Professor Martin Engelhard for discussions and several specific suggestions on the manuscript. RN acknowledges funding from the European Commission Marie Curie Training Network X-Probe, the Swedish Research Council (grant No. 2015-00560) and the European Research Council (ERC) under the European Union's Horizon 2020 research and innovation programme (grant agreement No 789030). GB acknowledges funding from the Swedish Research Council (grant 2017-06734).

## Author contributions
R.N. conceived the experiment, which was designed with input from R.B., G.B., J.S., D.J., F.D. and T.T. R.B., G.O., S.G. and T.B.U. produced the sample. R.B., G.O. and L.O. grew the *Np*SRII microcrystals used in these studies. R.B., G.O., S.G., D.J., P.N., G.H., T.B.U., T.W., F.D., T.T., J.S., G.B. and R.N. participated in data-collection. R.N., G.O. and L.O. participated in spectroscopic measurements. R.B. and G.O. processed and analysed the crystallographic data with support from T.W. R.N. and G.O. performed and analysed UV/vis spectroscopy measurements. R.N., R.B. and G.O. wrote the paper, with additional input from all authors.

## Funding

## Competing interests
The authors declare no competing interests.
