## [Transparent Peer Review file · Nature Communications]

Structural basis for the prolonged photocycle of sensory rhodopsin II revealed by serial synchrotron crystallography

Corresponding Author: Professor Richard Neutze

Editorial Note: Parts of this Peer Review File have been redacted as indicated to remove third party material where no permission to publish were obtained.

Version 0:

Reviewer comments:

Reviewer #1

(Remarks to the Author)

The manuscript reports results of serial synchrotron crystallography experiments at room temperature. Structural basis for a prolonged duration of the photocycle in sensory rhodopsin II (SRII) is revealed. By comparing the light-induced structural changes in SRII to earlier similar studies of bacteriorhodopsin (bR), the manuscript provides a highly significant insight into how relative flexibility of a helix controls a large difference in the duration of the photocycle between two molecules. This difference is directly related to different functions of these two proteins with very similar overall structures.

The manuscript makes important contribution to studies of proteins as dynamic systems that often reveal subtle changes in structure and flexibility that can potentially have a large impact on function. Dynamic studies are therefore essential and they are at the frontier of structural biology today. Depending on the time scale of interest, they can be done both at the ultra-high intensity XFELs and at synchrotrons X-ray sources, as this manuscript illustrates.

I recommend the manuscript for publication with only minor questions and suggestions listed below.

=====

p. 2, par. 2, line 6: "cis to trans" should be "trans to cis"

p. 5, par 1, line 4: Are these room temperature cell parameters. Should be stated.

p. 6, line 2: Table 1 is referred to in describing how 25 bins of 10ms duration were combined into 7 longer time windows. However, Table 1 only shows data collection and refinement statistics for the dark and continuous illumination data.

Remove reference to Table 1.

p. 8, line 1: Provide reference regarding "continuous illumination of pR crystals".

p. 9, par. 3, line 5: There is a reference to Figure 5d regarding helix C motion in SRII. Figure 5d doesn't show helices.

Explain or remove reference to Figure 5d.

p. 15, end of the paragraph: More details should be provided about refinement of the continuous-illumination SRII state. Was this a difference refinement (Terwilliger, T. C., and Berendzen, J. (1995) Acta Crystallogr. D Biol. Crystallogr. 51, 609–618)? Figure 1 caption: Arg75 mentioned but there is no Arg75 in the figure. Should be Asp75.

Figure 2: This figure is not really essential, can be moved to Supplementary Materials.

Figure 3 caption: Provide reference for bR steady state data.

Figure 4: It would be beneficial to add labels bR and SRII directly to figure panels a) and b) respectively, both in this figure and in figures 1, 5 and 6. That would make immediately apparent to the reader what panels correspond to, without a need to check the figure captions. It might also be beneficial to explore adding two more panels in this figure that show overlapped refined dark and steady state structures (without difference maps) for both bR and SRII. This would provide a view of overall structural changes throughout two molecules, while the important details are shown in Figures 5 and 6.

Figure 5: Display bR and SRII labels directly in the figure panels. Panel c: hydrogen bond Thr89-Asp85 shown intact, while text reports it is broken. Panel d: hydrogen bond Thr79-Asp75 not shown, while text reports it is intact. Model coloring: dark structure is shown in different colors in a) and c). It is shown as purple in a) and gray in c) while steady state structure is shown in purple in c). I suggest keeping the color of the dark state structure consistent in a) and c), distinct from the steady state color in c). Same for b) and d).

Figure 6: Display bR and SRII labels directly in the figure panels. a): label Wat404. Colors for helices for dark and steady state structures in c) and d) are too similar. Model coloring for side chains and retinal for dark and steady state structures: same comment as for Fig. 5.

Reviewer #2

(Remarks to the Author)

Sensory rhodopsin II (SRII) is a member of microbial rhodopsin superfamily, and it functions as a photoreceptor using a retinal chromophore for the negative phototaxis response of extremely halophilic archaea against blue light. Although the structure and amino acid sequence of SRII is similar to proton pumping rhodopsin, bR, it is not well understood how these different functions are exhibited from the common compact architectures. In this article, Bosman et al. conducted time-resolved serial synchrotron crystallography (TR-SSX) of SRII and succeeded in observing the structural change induced by light absorption. Furthermore, they compared the result with the result of bR observed by a similar method (Weinert et al. Science 2019). They found that there is a difference in the degrees of helical movement between SRII and bR. The smaller movement of SRII and the remaining interhelical hydrogen bond bridged by a water molecule can be related to the slower turnover rate of SRII photocycle which is important for their physiological function. Their experimental results are substantial. However, this reviewer needs to raise several scientific concerns about the significance and interpretation of their results as below.

One of the main results of this article is the smaller structural change of helices F and G compared with bR, keeping the hydrogen-bonding network between these helices. However, the active state of SRII was also structurally analyzed previously by conventional X-ray crystallography (Gushchin et al. JMB 2011), and the structure of the active state having similar inter-helical hydrogen-bonding between helices F and G was reported. The result of the current work is consistent with the previous one, however, it is unclear what new finding has been obtained by their room-temperature structural study, which is the most important to consider the significance of their work.

In the abstract, the authors described that “we performed time-resolved serial synchrotron crystallography (TR-SSX) studies...”, but it sounds to be somewhat misleading. The word “time-resolved” is generally used for the study which can trace the time-evolution of the photo-chemical event after a short stimulus (e.g. light). In contrast, the main structural insights of this article were obtained by using continuous laser illumination resulting in the observation of the photo-steady state. However, there is no description of this point in the abstract. This would raise some discrepancy between the impression by readers who read the abstract and the main results shown in the main text. To avoid the discrepancy, a description that the main results were obtained by continuous light illumination should be added to the abstract.

The description of how much protein was converted to the photo-intermediate by laser illumination was not clearly shown. Is there no possibility that the low amount of photo-converted molecule is the origin of the smaller Fo(light)-Fo(dark) signal compared with bR? A quantitative explanation about the fraction of photo-converted molecules in microcrystals is needed in the main text. Also, it is unclear that what photo-intermediate they observed. In the photocycle of SRII, there is two long-lived photo-intermediates M and O whose lifetimes are 150-700ms and ~400 ms. The continuous illumination would accumulate a mixture of both states. However, there is no description about how much amount of each intermediate was accumulated under their experimental condition and how they are related to each photo-intermediate with the solved structure. The assignment of photo-intermediates needs to be demonstrated in the text with substantial experimental evidence, and, if both M and O significantly contributed to the structure, heterogeneity between these intermediates should be taken into consideration during the structural modeling.

Minor points

Page 1, Abstract, 2nd line

Although two types of ion transporting microbial rhodopsins, active ion pumps and passive ion channels, are known, only the former was mentioned. The passive ion channels should be included here.

Page 2, Introduction, 1st paragraph

Although the authors listed various types of rhodopsins, many important proteins were lacking.

e.g.

proton pump: Leptosphaeria rhodopsin, xanthorhodopsin

inward proton pump: xenorhodopsin, schizorhodopsin

ion pump rhodopsin: NaR (Na⁺ pump), CIR (Cl⁻ pump)

hyperpolarization: anion channelrhodopsin

It would be better to include these comparatively new members of microbial rhodopsins. Also, this reviewer would like to show one of the most recent reviews on the sub-families of microbial rhodopsins: Rozenberg et al. Annu. Rev. Microbiol. (2021))

Page 2, Introduction, 1st paragraph 6th line

“cis to trans” must be “trans to cis”

Page 13, 1st paragraph

How much column-volume wash buffer was applied for washing the column?

Page 13, 2nd paragraph

“CaCl” must be “CaCl₂”

Figure 2, caption, 3rd line

Should (blue) be (orange)?

Figure 1

Only here, sensory rhodopsin II is written as "NpSRII". For consistency with the main text, it would be better to be changed to "SRII".

Reviewer #3

(Remarks to the Author)

Review report for manuscript NCOMMS-21-47815

"Structural basis for the prolonged photocycle of sensory rhodopsin II revealed by serial synchrotron crystallography", by R. Bosman et al.

General comments:

The manuscript reports a comparative study of the light-induced structural changes in two retinal proteins, namely bacteriorhodopsin (bR) and sensory rhodopsin II (sRII), with the aim being to provide a rationale for the reported slower photocycle, in particular the reprotonation step (M2-to-O) in sRII as compared to bR. The manuscript identifies indeed important differences, such as smaller structural changes in helices C, F and G, in sRII, related to the fact that this protein is not a proton pump. The study relies on a time-averaged experiment, i.e. it does not provide time-resolved information on these structural changes. But it makes sense that they are associated with the long-lived intermediates such as M2 and O; however both states cannot be differentiated.

The reported data are new for sRII and the question addressed is of significance for the community in the area of "photo-sensitive proteins" at large. These two criteria are a condition for publishing in almost all scientific journals. I do not see a particular reason that would justify publication in a high impact journal though, such as Nature Comm. One cannot qualify this work as a particular breakthrough. The reader expects to see information on how reprotonation occurs in sRII based on the particular motions of helices F and G.

From a technical perspective, a new crystallization method for sRII is reported. Recently, time-resolved VIS-pump/X-ray probe experiments were published in the notorious high impact journals, when time-resolved data were reported. This is not the case here. I would recommend publishing in a more specific journal with the relevant readership in the area of structural biology.

Discussion of results and conclusions:

The conclusions are sufficiently supported by the experimental results, as far as the photo-induced changes are concerned. Some points, however, appeared to be unclear or contradictory as detailed below.

- P.8: Sentence "Moreover, the same pattern of negative electron density features on neighbouring water molecules (WatA5 and WatA19),..." assign the positive electron density to "a new water molecule (WatB4)". In which sense is this molecule "new", rather than a relocation of one of the existing ones? And why should this signal be due to water and not the another side chain moving?

- In the following, it would be good to clarify for a non-expert to which helix Asp75 belongs. Which part of the top view changes (fig. 4b) are due to the very significant motion of Asp75?

- P. 8: "Negative difference electron density indicating the disruption of the Asp85-Thr89 H-bond in bR...". OK, but why does fig. 5C show a dashed line indicating a maintained H-bond?

- On p. 9, it is indicated that the Asp75-Thr79 H-bond is preserved in sRII, but the dashed line is removed in fig. 5D.

- On the same page, the sentence "This synchrony infers that the loss of this H-bond after proton-transfer to Asp85 prevents a futile back-reaction that would otherwise reprotonate the SB" is not clear. Do you mean a back reaction, i.e. reprotonation of the SB through Thr 79? And why is the back reaction from Asp85 unlikely?

Flaws in presentation of data and their interpretation:

Timing under microsecond pumping: The section regarding the photo-activation of the crystals by the ms pulses are unclear and difficult to follow (p.5). Instead of referring to the method section, it would be better to present here the important data such as crystal size, average distance between crystals, the velocity of the jet in terms of microns per sec as well as the nominal sizes (diameter=FWHM of intensity profile) of both laser and X-ray beam spots. The meaning of the 25 sequential bins is not clear for a non-expert, and it would be very beneficial to have a schematic graph with the laser pulse sequence and the X-ray shots.

It is in particular not clear at which rate the crystal arrive at the beam focus. In average one crystal per laser flash?

Another schematic should show that the X-ray beam (5 microns vertically) probes different regions of the homogeneously illuminated crystal as it advances. This will make clear that for an average crystal size of 60 microns, only 7-8 X-ray flashes can probe the crystal during its transit time, assuming that the first X-ray flash hit the lower crystal edge. This means a 70-80 microsecond transit time. These details are needed to understand the following discussion about the origin of relatively long-lasting diff. X-ray signals.

Light scattering in a LCP/monoolein mixture: An spectroscopic and quantitative characterization of light scattering was reported in ref. 34, with the conclusion that this effect is negligible. Indeed an LCP/monoolein mixture appears almost clear to the human eye. Hence, when the authors claim that a 28 micron spot radius can be enlarged to 90 microns, they should provide experimental evidence for it. I mean an independent experiment, in which the focus is monitored and compared between a water jet and a LCP/monoolein jet.

I suppose that the spatial overlap of laser and X-ray beam foci was carefully and reliably checked and that it was made sure that both foci coincide with the micro-jet position. It would be good to know how they did this, since this is not a trivial experiment.

But, the reason for the unexpectedly long-lived diff. X-ray signals is a different one: the assumption that photo-activation

would occur only in the 56 micron FWHM of the laser intensity profile is wrong. The intensity profile is not a square function, but a 2D Gaussian. There is no cut-off at the beam radius. The question is "at which distance from the beam center is the laser excitation negligible?". Given that the excitation factor ($\sigma F/h\nu$) is 66 in this experiment, the number of photons at a distance of 90 microns from the beam center is still $\gg 1$. In other words, crystals are highly excited and partially photo-converted without any problem 150 microseconds before crossing the X-ray beam.

I suggest the authors check this by running a series of laser power dependent experiments. This would make much more sense than the strange claim of a clear jet matrix inducing long range scattering over tens of microns.

In conclusion, besides the above technical flaws, the paper is sound and reports interesting new results explaining the different time scales of the photo-cycles in bR and sRII. However, before publication in any journal, the above problems have to be solved. In particular, regarding the scattering issue by the jet matrix, the present manuscripts adds to the confusion existing in the literature. New claims about the scattering properties, beyond those published in ref. 39, should be based on separate control experiments as suggested above.

And last, I do not see a particular a clear-cut argument that would qualify the present manuscript for publication in Nature Comm. But this is of course, the editor's decision.

Version 1:

Reviewer comments:

Reviewer #1

(Remarks to the Author)

Nature Communications revised manuscript NCOMMS-21-47815

I am satisfied with authors' responses to questions and suggestions from my review of the original manuscript. I recommend publication in Nature Communications.

Minor comments:

- Fig 3a: 0 to 30ms listed in the figure and figure caption but 30 to 60ms listed in the text (p. 8)
- Extended Fig 2a: "Laser illumination 10ms" listed in the figure, while 5ms laser burst listed in the figure caption and in the text (p. 7 and 19).
- p. 11, second paragraph: "a new water molecule (WatB4)" Should this be WatB5?

Reviewer #2

(Remarks to the Author)

The authors have appropriately addressed the reviewer's comments and have revised the descriptions in their manuscript accordingly, which has made their work suitable for publication in Nat. Commun. In particular, the new results of the spectroscopy on the microcrystals, which were used to identify the photointermediate species accumulated during the structural analysis, have significantly enabled a correlation between the observed structural changes and the chemical events occurring during the photocycle. I appreciate the authors effort and strongly recommend accepting this manuscript for publication.

Reviewer #4

(Remarks to the Author)

The manuscript by Bosman et al describes a detailed crystallographic study on sensory rhodopsin (NpSRII) aimed at understanding the structural basis of its dramatically slowed photocycle compared to bacteriorhodopsin (bR). The difference in lifetime of the last steps in the photocycle is functionally important: It affords a fast turnover of bR (a proton pump creating the proton gradient driving the ATPase, thus high proton flux rate is important) whereas the much longer lifetime of the active state of the blue light sensor SRII sets the stage for the signaling cascade. The manuscript is interesting, the experimental approach well described. Despite the fact that manuscript already underwent one cycle of revision, another one is needed. My comments are below, as they came up when reading the manuscript.

Similar to previously published studies on bR the manuscript describes serial synchrotron crystallography (SSX) experiments. To this end, SRII microcrystals were injected in a LCP stream into the X-ray beam. Jet speed does not seem to have been measured but calculated based on the flow rate of 0.2 ul/min. During injection, the crystals were either kept in the dark or illuminated using a 488 nm laser diode, focused to a $\sim 75 \times 75$ um (FWHM) spot, centered on the X-ray beam position. The laser energy is quite high; $F\sigma/h\nu$ is given as 35 (page 8) or 66 (page 19) [I did not check which one is correct]. In addition to light-induced structural changes, the heating of the protein upon absorption of a large number of photons may be the reason for – or at least contribute to - the observed reduction of diffraction resolution of illuminated crystals. The statement "Although microcrystals will be heated by some extend ... during continuous illumination these crystals continue to diffract, demonstrating that the gas stream surrounding an [should be the] LCP microjet provided effective cooling" misses the point. Supporting significant heating effects, the average B-factor of the continuously illuminated data is much higher than that of all other structures. It would be interesting to compare Wilson B factors. For the time-resolved SSX experiment, the laser and detector were triggered via a TTL pulse giving a 5 ns laser pulse. 25 X-ray exposures of 10 ms were collected before the arrival of the next laser pulse (4Hz). The time-resolved data were binned

into 30 ms.

Analysis of the light/dark difference density as a function of the X-ray pulse number after the pump laser flash showed good and internally consistent signal for 12-15 pulses or 120-150 ms. This duration also set the nominal exposure time of the continuous illumination dataset, which however is given as 120 ms instead of 150 ms. I agree, it should be 120 ms.

Steady state is a state or condition of a system that does not change in time. This implies that the system has undergone several cycles and that the observation time is longer than longest time-constant. Otherwise, the occupancy of the various states is still changing. It thus not correct to refer to a 120 ms exposure as steady state illumination, it is a photostationary state which is different from the steady state of the system.

Why was the spectral evolution not measured with crystals in LCP? (see below on implications for the occupancy). The beauty of the presented difference method is that no baseline/reference is needed which is very difficult to measure with LCP samples. It should be stated explicitly how "basis spectra for the photo-intermediates were extracted from the time-dependent sequence ... and the relative populations of the photo-excited M and O intermediates were quantified" (page 7); this information is missing in the methods part. [BTW, the M and O states are not photoexcited, simply delete this from the sentence.]

Determining the occupancy of intermediates is often challenging. Unfortunately, it can have major impact on the magnitude of the structural changes in the intermediate derived by refinement. The photocycle of SR11 in LCP is much faster than published previously; it is thus quite conceivable that the crystallized protein differs from both. How robust is the assumption of equal kinetics (and thus intermediate occupancies) of the soluble versus crystallized protein in LCP? Was the occupancy value checked for example with Xtrapol8 (De Zwitter et al)? Could it be that it is not noise that results in "loss of contrast" during partial occupancy refinement but using the wrong occupancy?

In view of the authors' comments on their refined structures, basically saying that they are not in line with the light-dark difference electron features (Fig. 4b, page 9), I find it quite surprising that the refined structures are used to back up interpretation of small structural changes (page 12).

Writing "Helix E/F" in Fig. 4c is misleading, according to the main text "the strongest electron density changes are associated with helices C, F and the extracellular loop linking helices D and E" (page 9). Please add error bars to fig. 4c, it seems that the temporal behavior of the difference electron density of Helix C and Helix E/F are highly similar and not "distinct". Without error bars (obtained for example by resampling techniques) it seems like a stretch to interpret the slightly different behavior in terms of different kinetics and function (entire first paragraph on page 10). The statement "...movement of helix C decays more rapidly than that of helix F is also implied from [should be consistent with] the steady-state observations (Fig. 4a,c) since the difference electron density features are more pronounced than [should be than] those associated with the F helix during continuous illumination (Fig. 3a-c)" middle of first paragraph page 10. Without a plot of the magnitude of the difference electron density vs residue number I find this difficult to accept. What is striking is that there is a lot of green density along the F helix in the photostationary state (3c) but almost no blue density. This is unexpected for a dataset that has good SNR. Why is that? (Similarly, Fig. 3a has more blue than green density for the C helix.) The blue/green ratio is important, it suggests the direction of changes; something goes from here to there. They authors make a strong point that this correlation of negative and positive density is more reliable than structural refinement, so this should be commented on.

The time-dependence of mean amplitude of difference electron density in Fig. 4 b, c. should not be given in Angstroms

Conclusions:

In view of the significantly faster decay of the M and O intermediates of SR11 embedded in LCP than in solution and the slower decay of the intermediates in bR crystals than in bR solution, it seems challenging to correlate the magnitude of the structural changes observed in crystals with those of proteins in solution. Moreover, unlike in bR, in SR11 the F-helix and the EF loop are involved in crystal contacts.

Tyr199 forms a strong crystal contact in SR11. It is unclear whether this is the reason that the G helix of NpSR11 does not unwind or whether it is the undoubtedly important H-bond Try174-Thr204, or a combination of both.

In view of these crystal contacts, it would seem that the statement "Whether or not the difference in the amplitude of these motions observed for bR and NpSR11 is physiological, or due to differences in crystal packing, cannot be determined from the TR-SSX data alone" (page 14 top) is too optimistic.

Other comments:

Page 14: I do not think that it is the "unwinding of the G helix in bR that frees the carbonyl oxygen of Lys216 from its H-bond interaction to the amide nitrogen of Gly220, providing a site for a [grammar problem, check] water molecule (Wat453) to transiently bind" but simply the conformation of Lys216.

Presumably, literature values of the time constants are given in Fig. 1, the references should be added to the figure legend 1c.

Page 6:

"Whereas the high multiplicity of SSX data can sometime reveal unresolved electron density in the loops ...". This is not correct. High multiplicity is necessary but not sufficient, it affects data quality. SSX data quality cannot be better than that of good rotation data. The "extra" electron density sometimes visible in SSX electron density maps originates from differences in structural distributions at ambient and cryogenic temperature or from the influence of cryoprotectants.

I obtain an average velocity of 780 um/s for the jet speed.

Page 12 middle

“density features associated with the Asp75 and Thr89 side-chains of NpSRII” -> should be Thr79

Page 16
Tyr204-> Thr204

PDB codes have changed and should be updated. In the dark structure, Wat 410 is cornered by positive and negative difference electron density, suggesting that it should move closer towards Asp75 and Lys205. Looking at the two structures and their densities accessible in the PDB there is still quite a bit of difference electron density in the active site.

Chloride in the dark structure is replaced by a water in the “steady state” structure, why?

Crystals. Description is inconsistent. Main text, p5 bottom: “... grew as needles with the longest dimension typically 100 um or larger, and the smallest dimension less than 10 um, but the longer needles tended to break during handling”. Methods, p17: “These setups yielded crystals 40-60 um in their largest dimensions, which were suitable for SSX”. A micrograph showing the crystals should be added to the Supplement.

Extended Data Fig. 2. Laser illumination should be 5 ms not 10 ms.

It would be useful to add residue numbers to Fig. 4a, showing a direct connection between helices and amino acid regions.

Version 2:

Reviewer comments:

Reviewer #4

(Remarks to the Author): **See below**

(Remarks on code availability)

Reviewer #5

(Remarks to the Author)

My specialty is time-resolved spectroscopy. Therefore, I read the manuscript of the paper more carefully, especially the part about time-resolved absorption spectra. As far as the time-resolved spectral data are concerned, I think the authors have adequately responded to the comments of reviewer 4 and the manuscript has been appropriately revised. I do not find any problems in the revised manuscript.

(Remarks on code availability)

Reviewer #6

(Remarks to the Author)

The authors report a time-resolved serial synchrotron X-ray crystallography (SSX) study of *Natronomonas pharaonis* rhodopsin II (NpSRII), observing phototriggered conformational changes on the timescale of 10s to 100s of milliseconds. The motivating biological question is to determine how related rhodopsins alter the structural and dynamical consequences of retinal photoisomerization to accomplish diverse downstream outcomes, from vision to ion pumping. The motivating technical goal is to determine how effectively SSX can be used to perform longer timescale time-resolved crystallography studies in order to broaden access to the technique by making it less dependent on X-ray free electron laser (XFEL) resources. In summary, I think this is an interesting and well-performed study.

I note that I was not involved in the initial review of the manuscript and therefore I considered the revised document in light of the responses in the text and the authors' rebuttal. The most substantive of the several revisions appears to be the inclusion of spectroscopic data that supports the authors' structural conclusions. I note that these experiments are non-trivial to perform and analyze-the authors have clearly made a serious, good-faith effort to address prior critiques. I also agree with Reviewer 2 that these data add valuable information to the manuscript that allows a clear correlation to be made between crystallographically- and spectroscopically-observed intermediates. Reviewer 4 makes several comments about important technical issues that I feel that the authors have adequately addressed, particularly with regards to the correlation between kinetics of intermediates in crystallo vs in solution. It is well-established that there are differences in these rates. I concur with Reviewer 4 that any such discrepancy merits close consideration, however I believe that the authors have done all that they can to address this. Given the extensive prior review, my minor comments below are offered in the spirit of polishing edits/considerations for a manuscript that, in my opinion, reports important work of potentially broad interest to the time-resolved structural biology community.

Minor points:

Line 156: "they" is a typo.

Lines 187-192. The B-factor comparison is potentially informative, but I feel the more interesting observation is that the unit cell parameters for the continuously illuminated crystals are slightly larger than the others. This expansion is often observed during crystal heating (e.g. Wolff et al, PMID: 37723259) and may be the most relevant observation to make about potential heating. However, the other timepoints have identical cells, and perhaps these parameters were held constant during scaling. If so, that would weaken the argument that the cell of the continuously illuminated sample is meaningfully larger than others.

Lines 250-251 and elsewhere. Positive and negative peaks are mentioned and it might be helpful to mention what the sigma and $e^{-\Delta^3}$ values are for these peaks, particularly when comparative comments are being made about stronger vs. weaker features.

Line 289: "move in unison" indicates a specific correlated motion that is compatible with the data but not conclusively established by it. Perhaps this should be qualified with "consistent with"... or similar.

Line 357: What is the magnitude of the coordinate error referred to here? Is it Luzzati error, Cruickshank DPI, or something else? This should be clarified.

Lines 474-475. The sentence "Datasets were truncated using CCP4 TRUNCATE" is potentially confusing. I acknowledge that the program is called TRUNCATE, but crystallographers do not usually talk about truncating data so much as scaling, converting from intensities to amplitudes, etc. It would be better to explicitly say what TRUNCATE was used for here.

(Remarks on code availability)

Reviewer #4

I appreciate the author's efforts to address my comments, in particular not only by writing but also by performing additional experiments. While most of my previous concerns have been addressed, some remain open and new ones have come up. Without addressing these I cannot support publication of the manuscript, unfortunately.

Major points

The authors claim that “unlike bR, helix G of NPSRII does not unwind near the conserved lysine residue...and therefore transient water molecule binding sites do not arise immediately to the cytoplasmatic side of retinal. The structural differences prolong the duration of the NpSRII photocycle relative to bR...”. I am still not convinced whether the structural differences between the two proteins in their photo-stationary state are indicative of functional differences, due to crystal constraints or a combination of both. The authors believe it is the former. To address my objection that it is difficult to correlate functional states observed in crystallo with kinetic features derived in solution the authors analyzed the photocycle kinetics with the protein dispersed in LCP (before crystallization) and after crystallization by difference spectroscopy. I realize the difficulty of such an experiment and laud the authors for doing so. The procedure is now well described and yields very good results for the non-crystallized protein in LCP (Fig. 2 a,b). Due to the poor signal of the crystalline sample the authors used the basis spectra derived from the non-crystallized protein in LCP for the analysis of the spectra of the crystallized protein in LCP. This is far from ideal but probably necessary. The same analysis was applied, using the same kinetic scheme, fitting two exponential decays. However, in the case of the crystalline sample I do not agree with the approach/result. It seems that the M state decays (yellow line in Fig 2) but I doubt that the O state (blue line) is formed. This is apparent from both derived numbers (essentially same decay times for the M/O crossover populations and the decay of the photoexcited population). Occam's razor should be applied and the data should be fitted with the minimal scheme, that is one exponential decay. Then the residuals should be plotted. If they show an exponential like systematic decay, inclusion of the second exponential is warranted. If not, the second component is simply not there. If the latter is the case, the question arises whether the system can go through a functional photocycle in the crystal or whether the formation of the O state is simply so slow that it cannot be observed, at least within this observation window. More generally, with such kinetics, very slow formation of O followed by rapid decay, O will not accumulate and will be extremely hard (essentially impossible) to observe it with this experimental setup. Thus, my current conclusion of the analysis is that the data have been overfitted. This objection needs to be refuted. If I am not wrong, analysis needs to be repeated without any O state.

The statement by the authors concerning my comment relating to crystal contacts is not correct (“Reviewer #4 is mistaken. As shown in Nango et al., Science 2016 (Figure S6), the EF loop and the C-terminus of a symmetry related molecule form crystal contacts in the bR P63 LCP crystals.”); it starts with errors in Nango et al. Citing from Nango et al, Science 2016: “This discrepancy may be reconciled by noting that motions of these helices are severely restricted in the P6₃ crystal form because residues 165 and 166 of the E-F loop participate in crystal contacts with residues 232 and 234 of the C terminus (fig. S7). “

This statement is wrong, there are no crystal contacts between residues 165 and 166 of the EF loop and residues #232, 234. Residues 165/166 are close to a beta-turn from residues #70 to #74. It is unlikely that this closeness restricts the 165/166 movement, there is no H-bond or other strong interaction, just a van der Waals contact.

Concerning the C-terminus: Looking at 6rqp, the two C-terminal residues contact residues 129-131 of a symmetry-related ($X-Y, X, Z+1/2$) molecule, as well as residue 72 of the same symmetry-related molecule. In some of the other structures, such as 6g7h, the C-terminus is one residue longer; that residues then also contacts 128-129 on that same symmetry mate, as well as residues 63-64, the latter coming from $-X, -Y, Z+1/2$.

By contrast, as pointed out in my last review, in SR-II there are strong H-bonds between the backbone amide of Ser 154 and the backbone carbonyl of Gln#151 and between the backbone carbonyl of Glu151 and the backbone amide of Ser#154. This will definitely have an influence on mobility.

Moreover, as pointed out in the last review, there is a strong H-bond between Tyr199 and the backbone carbonyl of #Ala125. This may affect the structural changes of the F-helix, including changes around Thr204. In bR light-induced changes (Weinert et al) extend all

along the F-helix, including the region downstream of Val210 (corresponding to Tyr199 in SRII which is involved in the aforementioned crystal contact). In SRII the structural changes are generally smaller and strongly diminished in the helical turn around Ile197. Potentially the “anchoring” of Tyr199 could affect the structural changes of Tyr174/Asp201 (which are much larger in bR). Importantly, Tyr199 has been shown to form an essential H-bond with Asn74 (HtrII) in the complex between photosensor (SRII) and transducer (HtrII) (Gordelly, Nature 2002). Proton transport activity was shown to be blocked in SRI and latently in SRII by HtrI and HtrII binding, respectively (Schmies et al, 2001 PNAS, Sudo et al, Biophys J. 2001). Thus, “fixing” Tyr199 is worrisome.

In conclusion I do not agree with the argument of the authors (wrong as stated) that the two proteins face similar crystal contact restrictions. Whether this is the reason for the differences in structural changes/kinetics or the many amino acid substitutions (apart from the discussed Thr/Ala) is hard to say.

Diffraction data / Refinement

It should be indicated which dark data set belongs to which light-data set or what the difference is. Neither dark structure overlays perfectly with the dark structure in the illuminated crystal datasets, why is this?

The bond length between Retinal-C15 and Nz-Lys205 is 1.38Ang in the dark structure (9H1W) and 1.27 A in 8PWP (0-30 ms) and the other time-resolved structures. What is the reason?

Why was neither water nor retinal included in structure 9h1x?

Table S1

The number of unique reflections makes no sense. It appears as if the datasets in columns 3-8 are not merged since they contain roughly double the number of reflections as in column 1. But why is the number in column 2 so much lower than in column 1 despite the fact that the resolution is higher?

It would be good to refine all structures to the same resolution, in particular the low resolution. Why was a 10 Ang resolution cut chosen for the light-data? This should at least be explained in the methods section.

The number of diffraction hits/indexing rate should be changed for the datasets in columns 3-8. Clearly the ~ 800,00 diffraction hits refer to all the data which were later binned into the various time-slices such as 0-30 ms, 30-60 ms, ... The number of diffraction hits per time slice should be indicated (ca 7,500) and the indexing rate should refer to this subset. Indexed image -> indexed lattice.

The difference in unit cell length is quite big between the dark and illuminated crystals (I assume this is the photo-stationary dataset). I only checked structure 9h1x. At least in this case the unit cell constants given in the header differ from the ones given in table S1 as does the Wilson B factor. Please double check all structures/table S1.

Table S2:

I understand that Table S2 lists the spectroscopically determined populations of the various species. However, “structural refinement occupancies” is misleading since it implies to me

that this reflects the occupancies used for refinement. Clearly only two structures were refined/dataset, the dark population and an activated one, but not two activated ones. Please rewrite such that it becomes clear what is meant.

Minor points

Authors. This very likely needs modifications.

Takashi is the first name; in line with all the other authors it should be Takashi Tomizaki
I think the affiliations of Tomizaki Takashi and Florian Dworkowski are wrong. For sure Tomizaki should be

Paul Scherrer Institute / SLS or MX Team and not SwissFEL. Although Dworkowski has moved to SwissFEL recently it is highly likely that the experiment was performed when he was still a member of the MX group at the SLS.

Line 82: delete (i.e. a photostationary state) not appropriate here

Line 87: reveal how structural rearrangements (grammar issue, the sentence makes no sense)

Line 104: handling as -> handling when

Line 106: loading into the INJECTOR reservoir

Line 121 Calpha atoms relative to -> from

Line 138: correlated with A photostationary state

Line 156 after they microcrystals -> delete they

Line 189-192: I would assume this is due to the low data quality/multiplicity. You could test this by refining a dark dataset consisting of the same number of lattices as the time-resolved light data.

Line 334-335 "... This water molecule interacts with N ϵ of Trp171 (SRII) and O γ of Thr204 in *Np*SRII, and Trp182 (bR) and O γ of Ala215 in bR". Fix the errors, the water molecule has the same kind of interactions in both proteins. It interacts with the backbone carbonyl (**not** O γ) of Thr204/Ala215.

Line 469: Table 1 -> Table S1

Line 471: ... diffraction data (containing 835719 diffraction hits).

Point by point response to reviewers:

Reviewer's comment in blue, our response in black.

Reviewer #1 (Remarks to the Author):

The manuscript reports results of serial synchrotron crystallography experiments at room temperature. Structural basis for a prolonged duration of the photocycle in sensory rhodopsin II (SRII) is revealed. By comparing the light-induced structural changes in SRII to earlier similar studies of bacteriorhodopsin (bR), the manuscript provides a highly significant insight into how relative flexibility of a helix controls a large difference in the duration of the photocycle between two molecules. This difference is directly related to different functions of these two proteins with very similar overall structures.

The manuscript makes important contribution to studies of proteins as dynamic systems that often reveal subtle changes in structure and flexibility that can potentially have a large impact on function. Dynamic studies are therefore essential and they are at the frontier of structural biology today. Depending on the time scale of interest, they can be done both at the ultra-high intensity XFELs and at synchrotrons X-ray sources, as this manuscript illustrates.

I recommend the manuscript for publication with only minor questions and suggestions listed below.

We thank Reviewer #1 for the insight contained within this summary. In particular we appreciate the insight shown by Reviewer #1 in appreciating that comparative dynamics studies “are therefore essential and they are at the frontier of structural biology today”.

p. 2, par. 2, line 6: “cis to trans” should be “trans to cis”

Thank you for spotting this typo. It has been corrected.

p. 5, par 1, line 4: Are these room temperature cell parameters. Should be stated.

The SSX parameters are room-temperature, whereas data in reference 3 were collected at cryogenic temperature. This point is now explicitly stated.

p. 6, line 2: Table 1 is referred to in describing how 25 bins of 10ms duration were combined into 7 longer time windows. However, Table 1 only shows data collection and refinement statistics for the dark and continuous illumination data. Remove reference to Table 1.

We include a new figure, Extended data figure 2, to illustrate the experimental protocol. Moreover, we have updated Table 1 to include all data referred to in the manuscript, including five time-dependent structures (0 to 30 ms; 30 to 60 ms; 60 to 90 ms; 90 to 120 ms; 120 to 150 ms). The final three time-windows (150 to 180 ms; 180 to 210 ms; 210 to 250 ms) were judged to have too little signal to noise to warrant further structural analysis (Figure 4a).

p. 8, line 1: Provide reference regarding “continuous illumination of pR crystals”.

(note “bR”, not “pR”). The reference is Weinert *et al.*, Science 2019. It has been added.

p. 9, par. 3, line 5: There is a reference to Figure 5d regarding helix C motion in SRII. Figure 5d doesn't show helices. Explain or remove reference to Figure 5d.

The figures have been rearranged in light of the need to significantly modify the manuscript to address the concerns of other reviewers. In revising the manuscript, the connection between a motion of helix C and figure 5 is no longer made.

p. 15, end of the paragraph: More details should be provided about refinement of the continuous-illumination SRII state. Was this a difference refinement (Terwilliger, T. C., and Berendzen, J. (1995) *Acta Crystallogr. D Biol. Crystallogr.* 51, 609–618)?

We did not use difference refinement. Rather, we used partial occupancy refinement, following the protocol of Nango et al., *Science* 2016. We have added Table 2, which specifies both the occupancy of the photoexcited/resting species, and the all-trans/13-cis conformation of the retinal, which were extracted from the time-resolved spectroscopic data (Figure 2). Details on the partial occupancy refinement procedure are given on page 20 of the manuscript.

Figure 1 caption: Arg75 mentioned but there is no Arg75 in the figure. Should be Asp75.

Both Asp75 and Arg72 are now labelled in the figure.

Figure 2: This figure is not really essential, can be moved to Supplementary Materials.

The figure which shows the lipid molecules is now moved to become Extended data figure 1.

Figure 3 caption: Provide reference for bR steady state data.

Reference 10 (Weinert et al., *Science* 2019) is now cited in the figure caption.

Figure 4: It would be beneficial to add labels bR and SRII directly to figure panels a) and b) respectively, both in this figure and in figures 1, 5 and 6. That would make immediately apparent to the reader what panels correspond to, without a need to check the figure captions.

We have made this recommended change and added the labels.

It might also be beneficial to explore adding two more panels in this figure that show overlapped refined dark and steady state structures (without difference maps) for both bR and SRII. This would provide a view of overall structural changes throughout two molecules, while the important details are shown in Figures 5 and 6.

We have now included an additional figure, Extended data figure 6, which follows this request.

Figure 5: Display bR and SRII labels directly in the figure panels.

We have followed this suggestion. Thank you.

Panel c: hydrogen bond Thr89-Asp85 shown intact, while text reports it is broken.

We have corrected this panel. Thank you.

Panel d: hydrogen bond Thr79-Asp75 not shown, while text reports it is intact.

After further analysis, we have concluded that this H-bond is broken in the photoactivated state, and the figures have been drawn accordingly.

Model coloring: dark structure is shown in different colors in a) and c). It is shown as purple in a) and gray in c) while steady state structure is shown in purple in c). I suggest keeping the color of the dark state structure consistent in a) and c), distinct from the steady state color in c). Same for b) and d).

We thank the reviewer for this suggestion, but the alternative also creates problems since we then have to come up with a colour scheme to illustrate the different time-points. We have therefore stuck to the philosophy that we use purple for bR and orange for NpSRII, but where we have two structures superimposed we use colour to illustrate the structure we wish to emphasize (ie. the photoactivated structure) and white to illustrate the resting structure. Thus while we appreciate the sentiment of the reviewer, we feel that this provides the best presentation overall.

Figure 6: Display bR and SRII labels directly in the figure panels.

We have followed this suggestion.

a): label Wat404. Colors for helices for dark and steady state structures in c) and d) are too similar. Model coloring for side chains and retinal for dark and steady state structures: same comment as for Fig. 5.

We think that the colour scheme is ok as it is.

Reviewer #2 (Remarks to the Author):

Sensory rhodopsin II (SRII) is a member of microbial rhodopsin superfamily, and it functions as a photoreceptor using a retinal chromophore for the negative phototaxis response of extremely halophilic archaea against blue light. Although the structure and amino acid sequence of SRII is similar to proton pumping rhodopsin, bR, it is not well understood how these different functions are exhibited from the common compact architectures. In this article, Bosman et al. conducted time-resolved serial synchrotron crystallography (TR-SSX) of SRII and succeeded in observing the structural change induced by light absorption. Furthermore, they compared the result with the result of bR observed by a similar method (Weinert et al. Science 2019). They found that there is a difference in the degrees of helical movement between SRII and bR. The smaller movement of SRII and the remaining interhelical hydrogen bond bridged by a water molecule can be related to the slower turnover rate of SRII photocycle which is important for their physiological function. Their experimental results are substantial.

We thank reviewer #2 for this positive assessment.

However, this reviewer needs to raise several scientific concerns about the significance and interpretation of their results as below.

One of the main results of this article is the smaller structural change of helices F and G compared with bR, keeping the hydrogen-bonding network between these helices. However, the active state of SRII was also structurally analyzed previously by conventional X-ray crystallography (Gushchin et al. JMB 2011), and the structure of the active state having similar inter-helical hydrogen-bonding between helices F and G was reported. The result of the current work is consistent with the previous one, however, it is unclear what new finding has been obtained by their room-temperature structural study, which is the most important to consider the significance of their work.

Gushchin *et al.* collected conventional X-ray diffraction data to 2.6 Å resolution from a single crystal of SRII cooled to cryogenic-temperature. Their trapping protocol is described as:

“For the loading of the intermediate state, the crystals were illuminated with an argon–krypton ion laser at a wavelength of 488 nm (Omnichrome). Cryostream was blocked for 2 s. The illumination was turned off permanently after 1 s has passed after the recooling has started. X-ray data were collected in the dark.”

In that work the temperature was transient and unknowable and the time-resolution was no better than two seconds. Moreover, their trapping protocol caused the crystals to diffract to lower resolution than for the dark-state and their diffraction data became anisotropic, such that a final resolution of 2.6 Å had incomplete data in the outer shell. That article shows two figures to illustrate their structural conclusions: [REDACTED]

From the arrows drawn on the left-panel you can see that the authors suggested that helix G and helix A move downwards (towards the extracellular medium) and that helix F moves upwards (towards the cytoplasm). It is therefore not correct to suggest (as the reviewer implies) that these movements are seen in our data. Gushchin *et al.* did not show any $F_o(\text{light}) - F_o(\text{dark})$ isomorphous difference Fourier electron density maps, and therefore their structural conclusions are based upon models rather than the observed changes in electron density. Moreover, Gushchin *et al.* did not highlight specific comparisons between the structural changes observed for bR and *NpSRII*, and our manuscript has a completely different emphasis concerning how structural results are interpreted structurally, providing completely new insight into how evolution has selected for differences in structural dynamics to achieve function. We do not accept this assessment of the reviewer.

In our study, crystals were kept at room-temperature throughout all measurements, and the time-resolution of our work was 30 ms, and our $F_o(\text{light}) - F_o(\text{dark})$ isomorphous difference Fourier electron density maps are of excellent quality (Figures 3, 5, 6 & Extended data figure 4). The scientific merits of room-temperature time-resolved X-ray diffraction studies are widely accepted and are well illustrated by bacteriorhodopsin (bR). In particular, despite a very large body of work using low-temperature trapping studies from 1999 and onwards (reviewed in Neutze *et al.*, *BBA* 2002; Wickstrand *et al.* *BBA* 2015), the editors and reviewers of *Science* (Nango *et al.*, *Science* 2016; Nogly *et al.*, *Science* 2018; Weinert *et al.* 2019) and *Nature Communications* (Nass Kovacs *et al.*, *Nature Comm.* 2019) recognized the unique structural insights that emerge from time-resolved serial crystallography studies performed at room temperature over and above what had previously been published using cryo-trapping studies. Once again, we do not accept the opinion of the reviewer.

In the abstract, the authors described that “we performed time-resolved serial synchrotron crystallography (TR-SSX) studies...”, but it sounds to be somewhat misleading. The word “time-resolved” is generally used for the study which can trace the time-evolution of the photo-chemical event after a short stimulus (e.g. light). In contrast, the main structural insights of this article were obtained by using continuous laser illumination resulting in the observation of the photo-steady state. However, there is no description of this point in the abstract. This would raise some discrepancy between the impression by readers who read the abstract and the main results shown in the main text. To avoid the discrepancy, a description that the main results were obtained by continuous light illumination should be added to the abstract.

We thank Reviewer #2 for this comment. In response, the abstract and the closing paragraph of the introduction have been changed to explicitly state that structural results derive from both time-resolved data and data from the photostationary state.

The description of how much protein was converted to the photo-intermediate by laser illumination was not clearly shown. Is there no possibility that the low amount of photo-converted molecule is the origin of the smaller $F_o(\text{light})-F_o(\text{dark})$ signal compared with bR?

As shown in the long-distance overviews of the difference Fourier electron density maps (Figure 3), the amplitude of the difference electron density features associated with helix C are quite comparable with those of bR, yet the features associated with helix F, and to an even larger extent with helix G, are much weaker. This is also clearly visible in panel a of Figure 4, where the amplitude of difference electron density features associated with helix C is quite comparable between *NpSRII* and bR, yet the features associated with helices F and G are much weaker in *NpSRII*. Thus, on a relative scale, we can be confident that the conclusion that helix F and G move less in *NpSRII* than in bR is a robust conclusion.

A quantitative explanation about the fraction of photo-converted molecules in microcrystals is needed in the main text. Also, it is unclear that what photo-intermediate they observed. In the photocycle of SRII, there is two long-lived photo-intermediates M and O whose lifetimes are 150-700ms and ~400 ms. The continuous illumination would accumulate a mixture of both states. However, there is no description about how much amount of each intermediate was accumulated

under their experimental condition and how they are related to each photo-intermediate with the solved structure. The assignment of photo-intermediates needs to be demonstrated in the text with substantial experimental evidence, and, if both M and O significantly contributed to the structure, heterogeneity between these intermediates should be taken into consideration during the structural modeling.

We thank Reviewer #2 for these comments. These concerns have led to significant changes in the manuscript. In particular, we have now recorded time-resolved UV/vis spectroscopy data from *NpSRII* in the crystallization setups (Figure 2) and we quantify the relative populations of the M and O intermediates for our TR-SSX data (Table 2). This information was incorporated into structural refinement with partial occupancy (Table 2). The values of the crystallographic occupancy of the photo-activated species that are extrapolated from the time-resolved spectroscopic data (falling from 37 % to 28 % from 0 to 30 ms and 120 to 150 ms) and those estimated for the photostationary state (44 %) are consistent with the quality of the difference Fourier electron density and other methods of estimating the crystallographic occupancy.

To understand how structure changes with time, we extract the time-dependence of the relative difference electron density changes associated with the extracellular side of helix C and the cytoplasmic side of helix F (Figure 4c, pasted below). What is apparent from the time-evolution of these electron density changes is that there is a different time-dependence for the structural changes associated with the extracellular portions of helix C and the structural changes associated with the cytoplasmic portions of helices E and F. Specifically, the movement of helices E/F is maximal for the initial time-delay and this motion decays steadily with time, whereas that associated with helix C peaks slightly later ($\Delta t = 30$ to 60 ms) and then also decays more rapidly. This illustrates how additional information is available from a room-temperature time-resolved X-ray diffraction study relative to earlier low-temperature trapping studies.

Minor points

Page 1, Abstract, 2nd line: Although two types of ion transporting microbial rhodopsins, active ion pumps and passive ion channels, are known, only the former was mentioned. The passive ion channels should be included here.

We have edited the abstract to explicitly mention both active and passive ion-transport functions.

Page 2, Introduction, 1st paragraph: Although the authors listed various types of rhodopsins, many important proteins were lacking. *e.g.* proton pump: *Leptosphaeria* rhodopsin, xanthorhodopsin inward proton pump: xenorhodopsin, schizorhodopsin; ion pump rhodopsin: NaR (Na⁺ pump), CIR (Cl⁻ pump); hyperpolarization: anion channelrhodopsin. It would be better to include these comparatively new members of microbial rhodopsins. Also, this reviewer would like to show one of

the most recent reviews on the sub-families of microbial rhodopsins: Rozenberg et al. Annu. Rev. Microbiol. (2021))

These examples of microbial rhodopsin are now given and the suggested reference is added.

Page 2, Introduction, 1st paragraph 6th line; “cis to trans” must be “trans to cis”

We thank Reviewer #2 for spotting this typo and we have made the correction.

Page13, 1st paragraph: How much column-volume wash buffer was applied for washing the column?

The column volume depends on the initial volume of cells used for each purification. For n grams of cells, we use n/2 ml of resin and that establishes the column volume. This detail is now included in the methods.

Page 13, 2nd paragraph: “CaCl” must be “CaCl₂”

Thank you, we made the correction

Figure 2, caption, 3rd line: Should (blue) be (orange)?

We thank Reviewer #2 for noting the ambiguity and have edited the figure caption (now Extended data figure 1).

Figure 1: Only here, sensory rhodopsin II is written as “NpSR_{II}”. For consistency with the main text, it would be better to be changed to “SR_{II}”.

Following advice from another, we decided to systematically change to *NpSR_{II}* throughout the document.

Reviewer #3 (Remarks to the Author):

Review report for manuscript NCOMMS-21-47815

“Structural basis for the prolonged photocycle of sensory rhodopsin II revealed by serial synchrotron crystallography”, by R. Bosman et al.

General comments:

The manuscript reports a comparative study of the light-induced structural changes in two retinal proteins, namely bacteriorhodopsin (bR) and sensory rhodopsin II (sRII), with the aim being to provide a rationale for the reported slower photo-cycle, in particular the reprotonation step (M2-to-O) in sRII as compared to bR. The manuscript identifies indeed important differences, such as smaller structural changes in helices C, F and G, in sRII, related to the fact that this protein is not a proton pump.

We thank Reviewer #3 for this summary.

The study relies on a time-averaged experiment, i.e. it does not provide time-resolved information on these structural changes. But it makes sense that they are associated with the long-lived intermediates such as M2 and O; however both states cannot be differentiated.

Because of this comment and similar comments by other reviewers, we have extensively revised the manuscript to place more emphasis on the time-resolved nature of our data. As explained above, we have now recorded time-resolved UV/vis spectroscopy data from *Np*sRII in the crystallization setups (Figure 2) and we quantify the relative populations of the M and O intermediates for our TR-SSX data (Table 2). This information was incorporated into structural refinement with partial occupancy (Table 2). The values of the crystallographic occupancy of the photo-activated species that are extrapolated from the time-resolved spectroscopic data (falling from 37 % to 28 % from 0 to 30 ms and 120 to 150 ms) and those estimated for the photostationary state (44 %) are consistent with the quality of the difference Fourier electron density and other methods of estimating the crystallographic occupancy.

The reported data are new for sRII and the question addressed is of significance for the community in the area of “photo-sensitive proteins” at large. These two criteria are a condition for publishing in almost all scientific journals. I do not see a particular reason that would justify publication in a high impact journal though, such as *Nature Comm*. One cannot qualify this work as a particular breakthrough. The reader expects to see information on how reprotonation occurs in sRII based on the particular motions of helices F and G.

As explained above, the scientific merits of room-temperature time-resolved X-ray diffraction studies are widely accepted and are well illustrated by bacteriorhodopsin (bR). In particular, despite a very large body of work using low-temperature trapping studies from 1999 and onwards (reviewed in Neutze *et al.*, *BBA* 2002; Wickstrand *et al.* *BBA* 2015), the editors and reviewers of *Science* (Nango *et al.*, *Science* 2016; Nogly *et al.*, *Science* 2018; Weinert *et al.* 2019) and *Nature Communications* (Nass Kovacs *et al.*, *Nature Comm.* 2019) recognized the unique structural insights that emerge from time-resolved serial crystallography studies performed at room temperature over and above what had previously been published using cryo-trapping studies.

For example, to understand how structure changes with time, we extract the time-dependence of the relative difference electron density changes associated with the extracellular side of helix C and the cytoplasmic side of helix F (Figure 4c, pasted below). What is apparent from the time-evolution of these electron density changes is that there is a different time-dependence for the structural changes associated with the extracellular portions of helix C and the structural changes associated with the cytoplasmic portions of helices E and F. Specifically, the movement of helices E/F is maximal for the initial time-delay and this motion decays steadily with time, whereas that associated with helix C peaks slightly later ($\Delta t = 30$ to 60 ms) and then also decays more rapidly. This illustrates how

additional information is available from a room-temperature time-resolved X-ray diffraction study relative to earlier low-temperature trapping studies.

From a technical perspective, a new crystallization method for sRII is reported. Recently, time-resolved VIS-pump/X-ray probe experiments were published in the notorious high impact journals, when time-resolved data were reported. This is not the case here. I would recommend publishing in a more specific journal with the relevant readership in the area of structural biology.

As explained above, our data are time-resolved. We have extensively revised our presentation of this work so as to emphasize the time-resolved nature of this study.

Discussion of results and conclusions:

The conclusions are sufficiently supported by the experimental results, as far as the photo-induced changes are concerned. Some points, however, appeared to be unclear or contradictory as detailed below.

- P.8: Sentence "Moreover, the same pattern of negative electron density features on neighbouring water molecules (WatA5 and WatA19),..." assign the positive electron density to "a new water molecule (WatB4)". In which sense is this molecule "new", rather than a relocation of one of the existing ones ? And why should this signal be due to water and not the another side chain moving ?

We do believe that the transient ordering of WatB4 arises from the reordering of one of the water molecules that are initially in the extracellular portion of the protein in the resting conformation. X-ray crystallography, however, cannot determine which of these water molecules is ordering.

- In the following, it would be good to clarify for a non-expert to which helix Asp75 belongs.

Asp75 belongs to helix C. This has been stated in the text at the first mention of this residue.

Which part of the top view changes (fig. 4b) are due to the very significant motion of Asp75 ?

We have added helix names to the figure so that it is easier for the reader to identify helix C, which is the helix to which Asp75 belongs.

- P. 8: "Negative difference electron density indicating the disruption of the Asp85-Thr89 H-bond in bR..". OK, but why does fig. 5C show a dashed line indicating a maintained H-bond ?

We have made this correction and the H-bond between Asp85 and Thr89 of bR is no longer indicated for the photoactivated state (now panel d of Figure 5).

- On p. 9, it is indicated that the Asp75-Thr79 H-bond is preserved in sRII, but the dashed line is removed in fig. 5D.

After further analysis, we have concluded that this H-bond is broken in the photoactivated state, and the figures have been drawn accordingly.

- On the same page, the sentence "This synchrony infers that the loss of this H-bond after proton-transfer to Asp85 prevents a futile back-reaction that would otherwise reprotonate the SB" is not clear. Do you mean a back reaction, i.e. reprotonation of the SB through Thr 79 ? And why is the back reaction from Asp85 unlikely ?

This sentence is rewritten to read:

"Moreover, in TR-SFX studies of bR, transient negative difference electron density was observed between Asp85 and Thr89, which illustrates how the Asp85-Thr89 H-bond is broken, arose as the SB became deprotonate.⁹ By breaking this connection once Asp85 was protonated, the pathway for reverse proton-exchange from Asp85 to the SB via Thr89 is disrupted and hence this futile back-reaction was prevented."

Flaws in presentation of data and their interpretation:

Timing under microsecond pumping: The section regarding the photo-activation of the crystals by the ms pulses are unclear and difficult to follow (p.5). Instead of referring to the method section, it would be better to present here the important data such as crystal size, average distance between crystals, the velocity of the jet in terms of microns per sec as well as the nominal sizes (diameter=FWHM of intensity profile) of both laser and X-ray beam spots. The meaning of the 25 sequential bins is not clear for a non-expert, and it would be very beneficial to have a schematic graph with the laser pulse sequence and the X-ray shots. It is in particular not clear at which rate the crystal arrive at the beam focus. In average one crystal per laser flash?

In response to this request, we include a schematic which illustrates how time-resolved data were collected (Extended data figure 2). The experimental setup is described as:

“Light-induced conformational changes in microcrystals of *Np*SRII were recorded both during continuous illumination (which equates to an approximate exposure of 120 ms due to the time required for the LCP jet to transit into the centre of the laser focus) and after microcrystals were exposed to blue-light flashes 5 ms in duration at a repetition rate of 4 Hz. An LCP flow-rate of 0.2 $\mu\text{l}/\text{min}$ using a 75 μm diameter LCP microjet allowed the sample to be translated approximately 190 μm between each 5 ms flash (*ie.* an average velocity of 760 $\mu\text{m}/\text{sec}$) and the same region of the LCP microjet was never exposed to blue light more than once.”

The pump laser spot size ($75 \times 75 \mu\text{m}^2$) and X-ray spot size ($20 \times 5 \mu\text{m}^2$) are given in the methods and the former is discussed in the text as appropriate. The indexing rate of about 7 % is given in the table of crystallographic data, so only a fraction of the images yielded diffraction data which could be processed. We also describe the dimensions of the microcrystals in the methods as:

“Microcrystals grew as needles with the longest dimension typically 100 μm or larger, and the smallest dimension less than 10 μm , but the longer needles tended to break during handling.

Another schematic should show that the X-ray beam (5 microns vertically) probes different regions of the homogeneously illuminated crystal as it advances. This will make clear that for an average crystal size of 60 microns, only 7-8 X-ray flashes can probe the crystal during its transit time, assuming that the first X-ray flash hit the lower crystal edge. This means a 70-80 microsecond transit time. These details are needed to understand the following discussion about the origin of relatively long-lasting diff. X-ray signals.

Since the LCP jet velocity was 760 μm per second, and the X-ray beam was only 5 μm high, then the transit time of any given microcrystal is set by the projection of that microcrystal in the vertical direction divided by the velocity. However, it is unclear why the reviewer is emphasizing this point, since it is the size of the light exposed region which is relevant. It doesn't matter in practice if any given microcrystal spanned this dimension, or was only partially in the illuminated region. The specific geometry for any given microcrystal is lost when determining the average X-ray diffraction intensity from thousands of microcrystals.

Light scattering in a LCP/monoolein mixture: An spectroscopic and quantitative characterization of light scattering was reported in ref. 34, with the conclusion that this effect is negligible. Indeed an LCP/monoolein mixture appears almost clear to the human eye. Hence, when the authors claim that a 28 micron spot radius can be enlarged to 90 microns, they should provide experimental evidence for it. I mean an independent experiment, in which the focus is monitored and compared between a water jet and a LCP/monoolein jet.

This is obvious from the photograph of the illuminated LCP microjet shown in Extended data figure 2 (which was included as Figure 3b in the previous submission).

I suppose that the spatial overlap of laser and X-ray beam foci was carefully and reliably checked and that it was made sure that both foci coincide with the micro-jet position. It would be good to know how they did this, since this is not a trivial experiment.

This procedure is now extremely standard and is described in the methods as:

“The X-ray beam position was located and aligned using a YAG screen (which is fluorescent when exposed to X-rays) and a high-magnification camera. The focal spot size of the laser diode was determined offline using a knife-edge scan. The laser diode was aligned on the extruding jet using an attenuated beam that allowed centering of the laser spot vertically by centering the maximal intensity on the X-ray beam. Horizontal alignment was aided by laser scattering from the LCP jet, being aligned to produce a symmetric speckle pattern behind the jet.”

But, the reason for the unexpectedly long-lived diff. X-ray signals is a different one: the assumption that photo-activation would occur only in the 56 micron FWHM of the laser intensity profile is wrong. The intensity profile is not a square function, but a 2D Gaussian. There is no cut-off at the beam radius. The question is “at which distance from the beam center is the laser excitation negligible?”. Given that the excitation factor ($\sigma F/h\nu$) is 66 in this experiment, the number of photons at a distance of 90 microns from the beam center is still $\gg 1$. In other words, crystals are highly excited and partially photo-converted without any problem 150 microseconds before crossing the X-ray beam.

We accept this argument of the reviewer and have incorporated this argument into a shortened description of the issue of the transit time of the photoactivated samples through the X-ray beam on page 9 of the revised manuscript.

I suggest the authors check this by running a series of laser power dependent experiments. This would make much more sense than the strange claim of a clear jet matrix inducing long range scattering over tens of microns.

As noted above, the argument of the reviewer is accepted and there is no need for additional experiments on this point.

In conclusion, besides the above technical flaws, the paper is sound and reports interesting new results explaining the different time scales of the photo-cycles in bR and sRII. However, before publication in any journal, the above problems have to be solved. In particular, regarding the scattering issue by the jet matrix, the present manuscripts adds to the confusion existing in the literature. New claims about the scattering properties, beyond those published in ref. 39, should be based on separate control experiments as suggested above.

The reviewer is referring to reference 34: Grunbein, M. L. et al. Illumination guidelines for ultrafast pump-probe experiments by serial femtosecond crystallography. *Nat Methods* 17, 681-684, doi:10.1038/s41592-020-0847-3 (2020). However, since this issue is not central to the major results of the paper, we have let the point go and we accept the above argument of the reviewer that the photo-illuminated region is larger than that defined by the nominal FWHM of the laser spot.

And last, I do not see a particular a clear-cut argument that would qualify the present manuscript for publication in *Nature Comm*. But this is of course, the editor's decision.

This is the first article to use time-resolved serial synchrotron X-ray crystallography to investigate how evolutionary differences in function can be optimized by utilizing differences in protein conformational dynamics. As such, the paper breaks new ground for the scientific scope of the field while presenting new structural results from time-resolved diffraction measurements performed at room-temperature.

Reviewer #1 (Remarks to the Author):

Nature Communications revised manuscript NCOMMS-21-47815

I am satisfied with authors' responses to questions and suggestions from my review of the original manuscript. I recommend publication in Nature Communications.

We thank Reviewer #1 for this very clear recommendation.

Minor comments:

- Fig 3a: 0 to 30ms listed in the figure and figure caption but 30 to 60ms listed in the text (p. 8)

We thank Reviewer #1 for spotting this. The figure and figure caption were correct. This was a case of a figure being swapped out, but the main body text not being updated accordingly

- Extended Fig 2a: "Laser illumination 10ms" listed in the figure, while 5ms laser burst listed in the figure caption and in the text (p. 7 and 19).

Reviewer #1 is correct – the figure was in error and the text was correct. We have corrected the figure to consistently indicate that samples were exposed to a 5ms laser burst.

- p. 11, second paragraph: "a new water molecule (WatB4)" Should this be WatB5?

Reviewer #1 is correct – this should be WatB5 and this has been changed in the text.

Reviewer #2 (Remarks to the Author):

The authors have appropriately addressed the reviewer's comments and have revised the descriptions in their manuscript accordingly, which has made their work suitable for publication in Nat. Commun. In particular, the new results of the spectroscopy on the microcrystals, which were used to identify the photointermediate species accumulated during the structural analysis, have significantly enabled a correlation between the observed structural changes and the chemical events occurring during the photocycle. I appreciate the authors effort and strongly recommend accepting this manuscript for publication.

We thank Reviewer #2 for this very positive recommendation.

Reviewer #4 (Remarks to the Author):

The manuscript by Bosman et al describes a detailed crystallographic study on sensory rhodopsin (NpSRII) aimed at understanding the structural basis of its dramatically slowed photocycle compared to bacteriorhodopsin (bR). The difference in lifetime of the last steps in the photocycle is functionally important: It affords a fast turnover of bR (a proton pump creating the proton gradient driving the ATPase, thus high proton flux rate is important) whereas the much longer lifetime of the active state of the blue light sensor SRII sets the stage for the signaling cascade. The manuscript is interesting, the experimental approach well described.

We thank Reviewer #4 for this summary.

Despite the fact that manuscript already underwent one cycle of revision, another one is needed. My comments are below, as they came up when reading the manuscript.

Similar to previously published studies on bR the manuscript describes serial synchrotron crystallography (SSX) experiments. To this end, SRII microcrystals were injected in a LCP stream into the X-ray beam. Jet speed does not seem to have been measured but calculated based on the flow rate of 0.2 $\mu\text{l}/\text{min}$.

This is correct. The calculated value represents an average over time determined from the flow-rate parameters and microjet diameter. When these data were collected, there were no tools available to monitor the flow. Some tools have been developed to measure the flow-rate in real time at SwissFEL, but their method is based upon the visibility of a ladder due to X-ray exposure induced damage within the microjet. Monochromatic synchrotron radiation does not cause the same level of X-ray damage as a single-shot XFEL beam, and this method cannot be applied at the synchrotron. It is possible that the sample flow rate fluctuates somewhat during the experiment.

In the previous version of the manuscript, on page 8, we wrote:

“An LCP flow-rate of 0.2 $\mu\text{l}/\text{min}$ using a 75 μm diameter LCP microjet allowed the sample to be translated approximately 190 μm between each 5 ms flash (*ie.* an average downwards velocity of 760 $\mu\text{m}/\text{sec}$) and the same region of the LCP microjet was never exposed to blue light more than once.”

In revising the manuscript, we have revised the phrase within the brackets to read:

“(the average downwards velocity is calculated to be 760 $\mu\text{m}/\text{sec}$)”.

During injection, the crystals were either kept in the dark or illuminated using a 488 nm laser diode, focused to a $\sim 75 \times 75 \mu\text{m}$ (FWHM) spot, centered on the X-ray beam position. The laser energy is quite high; $F\sigma/h\nu$ is given as 35 (page 8) or 66 (page 19) [I did not check which one is correct].

This discrepancy is a typo and we thank Reviewer #4 for spotting this error. The source of the mistake is that another experiment utilized a 10 ms laser pulse-train (rather than 5 ms used in this experiment) and an internal miscommunication led to this error. Rounding to two significant figures, the correct value of product $F\cdot\sigma/h\nu \sim 37$ is now used consistently throughout the revised manuscript. Note, this idealized value assumes no experimental losses, yet in reality this product could be much lower for microcrystals. However, there is no agreed approach to estimate these losses, and therefore rather than diverge into that debate we quote the idealized case. We have revised the manuscript by quoting the value on the incident surface and including the phrase “but assuming no experimental losses due to scattering”, to now read:

“A 5 ms exposure using a continuous 2.6 mW laser focused into a 75 μm FWHM (full width half maximum) diameter spot yields the dimensionless product $F\cdot\sigma/h\nu \sim 49$ when incident upon the microjet surface and assuming no experimental losses due to scattering, and this value is ~ 37 when averaged throughout the volume of a representative microcrystal, where F is the laser pulse fluence averaged across the FWHM focal spot, σ is the

absorption cross section of the resting state of *NpSRII*, and $h\nu$ is the product of Planck's constant with the frequency of the light."

In addition to light-induced structural changes, the heating of the protein upon absorption of a large number of photons may be the reason for – or at least contribute to - the observed reduction of diffraction resolution of illuminated crystals. The statement "Although microcrystals will be heated by some extent ... during continuous illumination these crystals continue to diffract, demonstrating that the gas stream surrounding an [should be the] LCP microjet provided effective cooling" misses the point. Supporting significant heating effects, the average B-factor of the continuously illuminated data is much higher than that of all other structures. It would be interesting to compare Wilson B factors.

We thank Reviewer #4 for this careful reading. In fact, the average B-factor during continuous illumination in Table S1 was a typo. We have cross-checked the pdb reports and corrected all values.

We have therefore revised the above sentence to now read:

"Although microcrystals will be heated to some extent by the energy of the absorbed light, during continuous illumination these crystals continued to diffract. Some heating induced disordering might be implied by the observation that the average B-factor is 54.0 Å² for the dark data-set and 60.0 Å² during continuous illumination (**Table S1**). However, the average B-factor for the time-resolved data sets from 0 to 150 ms vary from 45.0 Å² to 47.0 Å² (**Table S1**). Since these values are lower than the dark data-set, sample to sample variation in crystals may also explain this observation."

In Table S1 has also been revised to now also included the Wilson B-factors as requested. These do not show the same trend. So we think that this is not a systematic effect from which any strong conclusion can be drawn, but rather reflects sample to sample variations.

For the time-resolved SSX experiment, the laser and detector were triggered via a TTL pulse giving a 5 ns laser pulse. 25 X-ray exposures of 10 ms were collected before the arrival of the next laser pulse (4Hz). The time-resolved data were binned into 30 ms. Analysis of the light/dark difference density as a function of the X-ray pulse number after the pump laser flash showed good and internally consistent signal for 12-15 pulses or 120-150 ms. This duration also set the nominal exposure time of the continuous illumination dataset, which however is given as 120 ms instead of 150 ms. I agree, it should be 120 ms.

We appreciate that Reviewer #4 agrees with this choice. The logic for the choice of a 120 ms exposure when performing the spectroscopic characterization was explained in the methods, and we now have edited the main text to also read:

"A blue continuous laser diode ($\lambda = 473$ nm) was aligned onto this capillary and a millisecond shutter was used to create pump-laser pulses of 5 ms in duration for time-resolved recordings. Likewise, pump-laser pulses of 120 ms in duration were used to observe a difference spectrum that could be correlated with photostationary state, since it takes approximately 120 ms for the LCP jet to transit through the focused laser spot and into the X-ray beam position."

Steady state is a state or condition of a system that does not change in time. This implies that the system has undergone several cycles and that the observation time is longer than longest time-constant. Otherwise, the occupancy of the various states is still changing. It thus not correct to refer to a 120 ms exposure as steady state illumination, it is a photostationary state which is different from the steady state of the system.

We thank Reviewer #4 for noting this semantic error and we have replaced two occurrences of "steady-state" within the document with "photostationary state".

Why was the spectral evolution not measured with crystals in LCP? (see below on implications for the occupancy).

When microcrystals grow, an initially homogeneous sample becomes inhomogeneous, with the protein becoming concentrated in crystals and the LCP often becoming turbid or birefringent in the region where crystals grow. This leads to well-known distortions in spectra as some light passes through crystalline samples, some light passes through the LCP but does not pass through microcrystals, and some light is scattered by sample inhomogeneities, birefringent regions associated with lipidic phase transitions, *etc.* These phenomena can cause spectra from microcrystals to appear somewhat flattened, but more often than not they cannot be used for quantitative purposes.

In the previous submission of this manuscript, we allowed LCP crystallization setups to equilibrate for a few days, which meant that the protein was fully incorporated into the lipidic cubic phase environment and had the same pH, ionic strength *etc...* as for microcrystals. There may have been crystal nuclei by this point in time, but these samples appeared to be homogeneous, we got very good measurements from these samples, and the temporal evolution could be reliably extracted (Figure 2A, B, C). To reply in full to Reviewer #4, we grew and purified protein, grew microcrystals and recorded time-resolved spectroscopy data from microcrystalline samples. For the reasons described above, these data (Figure 2D, E, F) were not nearly as good as when using the homogeneous samples shortly after crystallization experiments were setup. Nevertheless, by using the basis spectra extracted from the earlier spectroscopic studies, we could fit these data and we observed that the transition from the M-to-O intermediate is slower in the crystalline state.

The section Spectral evolution of NpSR_{II} within LCP crystallization conditions on page 7 and 8 has been heavily revised to reflect these changes.

The beauty of the presented difference method is that no baseline/reference is needed which is very difficult to measure with LCP samples.

Reviewer #4 is correct. We thought this approach a good compromise, but now we also present spectra from microcrystals (Figure 2).

It should be stated explicitly how “basis spectra for the photo-intermediates were extracted from the time-dependent sequence ... and the relative populations of the photo-excited M and O intermediates were quantified” (page 7); this information is missing in the methods part. [BTW, the M and O states are not photoexcited, simply delete this from the sentence.]

We have edited the text on page 7 to now read:

“Basis spectra for the photo-intermediates of NpSR_{II} were extracted from the time-dependent sequence of difference absorption spectra (**Figure 2a**) and the relative populations of the M and O intermediates were quantified (**Figure 2b**) using a linear decomposition script which assumes an exponential decay of the first component into the second component, and of the second component back to the resting state. In this routine, two basis-spectra (for the M and O spectral intermediates, respectively) were optimized as a linear sum of the first two singular value decomposition (svd) components extracted from the set of difference spectra, as has been described in other contexts (section 1.3.3 of supplementary material of reference³⁵ provides details).³⁵”

Equations 3 and 4 of the supplementary material of reference 35 explain this procedure. Linear decomposition of a set of spectra is a very old technique and very standard, and we therefore don't think it is necessary to place more emphasis on this point. We have posted the Matlab script and data used to generate Figure 2 (left column) on Github at: https://github.com/Neutzelab/TR_SX_SRII.

Reviewer #4 is correct, we are referring to M and O intermediates, so we have deleted the word “photo-excited” in this context.

Determining the occupancy of intermediates is often challenging. Unfortunately, it can have major impact on the magnitude of the structural changes in the intermediate derived by refinement. The photocycle of SRII in LCP is much faster than published previously; it is thus quite conceivable that the crystallized protein differs from both. How robust is the assumption of equal kinetics (and thus intermediate occupancies) of the soluble versus crystallized protein in LCP?

As discussed above, we have revised the article with the addition of time-resolved spectroscopy data measured from microcrystals slurries (Figure 2). We agree with Reviewer #4 that determining the crystallographic occupancy of photoactivated species is challenging, and we agree that this affects the amplitudes of modelled motions. For this reason, we relied upon the more easily quantified results from our spectroscopic analysis.

Reviewer #4 writes “The photocycle of SRII in LCP is much faster than published previously”. In fact, there is considerable variation in the literature concerning the lifetime of M-state. For example, the photocycle below is modified from Figure 6 of K. Inoue et al. / *Biochimica et Biophysica Acta* 1837 (2014) 562–577. The M-state has a measured lifetime of 200 ms to 1700 ms and depends upon the experimental conditions. We are the first to measure this in LCP phase under the crystallization conditions of pH 7.5, CaCl₂ 150 mM, Glycine 100 mM, 38% (v/v) PEG 400. [REDACTED]

Reviewer #4 writes “it is thus quite conceivable that the crystallized protein differs from both. How robust is the assumption of equal kinetics (and thus intermediate occupancies) of the soluble versus crystallized protein in LCP?” To respond to this question, we have measured the kinetics in microcrystals. The decay of the M-state and growth of the O-state is slower than in the LCP setups, but is consistent with the photocycle pasted above.

Was the occupancy value checked for example with Xtrapol8 (De Zwitter et al)?

Following this recommendation of Reviewer #4, we used Xtrapol8 to analyse these data TR-SSX data. The resulting occupancy estimates extracted from Xtrapol8 are:

Δt (ms)	0 to 30	30 to 60	60 to 90	90 to 120	120 to 150	Continuous illumination
Occupancy	43 %	32 %	17 %	35 %	13 %	49 %

This gives an average occupancy for the four time-points with relatively strong difference electron density of 32 % \pm 11 %.

From the spectroscopy measurements we estimated (Table S2):

Δt (ms)	0 to 30	30 to 60	60 to 90	90 to 120	120 to 150	Continuous illumination
-----------------	---------	----------	----------	-----------	------------	-------------------------

Occupancy	37 %	35 %	33 %	31 %	28 %	44 %
-----------	------	------	------	------	------	------

Which gives an average for the first four time-points of $34 \% \pm 3 \%$. Thus, the spectroscopic and crystallographic estimates agree, but clearly the estimates from spectroscopy are more reliable than those from Xtrapol8. Likewise, the Xtrapol8 estimate for continuous illumination and that from spectroscopy are close (49 % vs 44 %) if one assumes that the uncertainty estimates are similar to the time-resolved estimates.

In revising the manuscript, we have edited **Table S2** to include this information. We have also modelled the decay of the overall crystallographic occupancy as the photoactivated sample is swept out of the X-ray beam. The difference that these details make to the results of structural refinement are very small.

Could it be that it is not noise that results in “loss of contrast” during partial occupancy refinement but using the wrong occupancy?

The short answer is “no”, this comment concerning “contrast” is not influenced by the choice of occupancy, which was correctly chosen.

It was a poor choice of words to write “loss of contrast” in the previous version of the manuscript. The “contrast” when quantifying the difference Fourier electron density maps seemed to be enhanced in when using the tool of Wickstrand *et al.* because we imposed a pedestal (of 2.0σ) for the difference density, below which the residual density is set to zero. This pedestal gave the appearance of enhanced contrast, but it was a choice. Moreover, the experimental data has noise, structural modelling is not perfect (*eg.* R-factors are typically the order of 15 % to 20 %), and chemical and energetic restraints imposed during structural modelling influence the modelled protein conformational changes. What we show is the correlation between the results of structural refinement and difference density maps. If Reviewer #4 were to use these tools to analyze all time-resolved X-ray crystallography studies they would find similar correlations, which are never perfect.

We recently showed (Vallejos *et al.*, *Structural Dynamics* 2024) that the amplitude of C α perturbations is inversely proportional to the chosen crystallographic occupancy. Therefore the amplitude of the modelled motions are affected if another occupancy were chosen, but the “contrast” (quantified by Pearson correlation function, see panels 1e and 1f of Vallejos *et al.*, *Structural Dynamics*, 2024) remains the same and does not depend upon the occupancy. Moreover, as shown above, both the crystallographic and spectroscopic estimates for the occupancy agree. The same article (Vallejos *et al.*, *Structural Dynamics* 2024) also showed that the results of structural refinement using partial occupancy refinement, and structural refinement against data extrapolated using Xtrapol8, agree within coordinate errors.

In view of the authors' comments on their refined structures, basically saying that they are not in line with the light-dark difference electron features (Fig. 4b, page 9), I find it quite surprising that the refined structures are used to back up interpretation of small structural changes (page 12).

This statement is false. Reviewer #4 is misrepresenting what we wrote. At no point did we say that the refined structures are not in line with the light-dark difference electron density features. In fact, we show how the results of structural refinement (Figure 4b) correlate with a one-dimensional representation of the difference Fourier map (Figure 4c). This correlation is not perfect because it can never be perfect. The coordinate representation is representing C α atom displacements, whereas the difference Fourier map analysis represents all atoms. Structural modelling must be consistent with chemical restraints, whereas there are no such restraints on difference Fourier maps. Coordinate errors and noise in difference Fourier electron density maps do not have an identical impact on these plots. If Reviewer #4 believes that these curves should be identical, then Reviewer #4 misunderstands what is being presented.

Writing “Helix E/F” in Fig. 4c is misleading, according to the main text “the strongest electron density changes are associated with helices C, F and the extracellular loop linking helices D and E” (page 9).

When redrawing this figure according to the recommendation of Reviewer #4 below. We have also changed the legend to say “Helix F” and not “Helix E/F”.

Please add error bars to fig. 4c, it seems that the temporal behavior of the difference electron density of Helix C and Helix E/F are highly similar and not “distinct”. Without error bars (obtained for example by resampling techniques) it seems like a stretch to interpret the slightly different behavior in terms of different kinetics and function (entire first paragraph on page 10).

We have followed the recommendation of Reviewer #4 and now utilize resampling methods to estimate error bars in Figure 4c. We also expanded this figure to include the time-dependent behaviour of features associated with the extracellular regions of helix D and E, and of helix G. Our resampling approach is based upon the method described in Vallejos *et al.*, *Structural Dynamics* 2024. With this addition, Reviewer #4 is correct that the decay of difference electron density on helix C and helix F show a very similar temporal trajectory.

The statement “...movement of helix C decays more rapidly than that of helix F is also implied from [should be consistent with] the steady-state observations (Fig. 4a,c) since the difference electron density features are more pronounced than [should be than] those associated with the F helix during continuous illumination (Fig. 3a-c)” middle of first paragraph page 10. Without a plot of the magnitude of the difference electron density vs residue number I find this difficult to accept.

The requested data was already shown in Figure 3a. But to better emphasise the point, we have edited Figure 4c to show both this analysis averaged over the time-delays $\Delta t = 0$ ms to 90 ms and the same analysis for data collected during continuous illumination.

The remainder of the paragraph has been so reworked that the suggested rewording (“consistent with”) is not used.

What is striking is that there is a lot of green density along the F helix in the photostationary state (3c) but almost no blue density.

This statement of Reviewer #4 is not correct. It can be seen from Figure 4a that the positive (blue) and negative (yellow) difference electron density are quite balanced for helix F. In fact, when we sum the positive or negative values shown in Fig. 4a, we observe that the positive and negative difference electron density features sum to within 2 % to 4 % of each other. There is no discrepancy.

This is unexpected for a dataset that has good SNR. Why is that? (Similarly, Fig. 3a has more blue than green density for the C helix.) The blue/green ratio is important, it suggests the direction of changes; something goes from here to there. They authors make a strong point that this correlation of negative and positive density is more reliable than structural refinement, so this should be commented on.

There is no problem. Reviewer #4 is mistaken. See comment above.

The time-dependence of mean amplitude of difference electron density in Fig. 4 b, c. should not be given in Angstroms

Reviewer #4 is correct, and the way the figure was drawn was misleading. We have therefore separated these data showing in Figure 4b into two separate plots, and corrected the Y-axis label in Figure 4c (now Figure 4D). We thank Reviewer #4 for spotting this.

Conclusions:

In view of the significantly faster decay of the M and O intermediates of SRII embedded in LCP than in solution and the slower decay of the intermediates in bR crystals than in bR solution, it seems

challenging to correlate the magnitude of the structural changes observed in crystals with those of proteins in solution.

One performs time-resolved X-ray diffraction studies on protein in crystals because without crystals there is no X-ray diffraction. For three decades, this well-established field of research has delivered numerous novel structural insights (reviewed in Bränden & Neutze, Science 2021). The community is well aware that the kinetics of any protein reaction are influenced by the environment, including whether or not they are in crystals.

As shown in Figure 1c, the rate constants in the literature for the SRII M-to-O transition differ by an order of magnitude (from 200 ms to 1.7 s, rate constants taken from Inoue et al., BBA 1837, 562 (2014)). We should point out that some crystal forms of bR have the kinetics slowed by a factor of 100 [Takeda et al., JMB 341, 1023, 2004]. For SRII, our data show that the overall photocycle is slowed over threefold when crystallized (Figure 2) and this means that the population of the O-state does not build up to the same extent in microcrystals.

Moreover, unlike in bR, in SRII the F-helix and the EF loop are involved in crystal contacts.

Reviewer #4 is mistaken. As shown in Nango et al., Science 2016 (Figure S6), the EF loop and the C-terminus of a symmetry related molecule form crystal contacts in the bR P6₃ LCP crystals.
[REDACTED]

For the SRII C222₁ LCP crystal form, the residues 151 and 154 of the EF-loop interact with residues 154 and 151 of a symmetry related molecule.

In view of these crystal contacts, it would seem that the statement “Whether or not the difference in the amplitude of these motions observed for bR and NpSRII is physiological, or due to differences in crystal packing, cannot be determined from the TR-SSX data alone” (page 14 top) is too optimistic.

We cannot respond to this comment since we do not understand what point Reviewer #4 is trying to make.

Other comments:

Page 14: I do not think that it is the “unwinding of the G helix in bR that frees the carbonyl oxygen of Lys216 from its H-bond interaction to the amide nitrogen of Gly220, providing a site for a [grammar problem, check] water molecule (Wat453) to transiently bind” but simply the conformation of Lys216.

Once again, we do not understand Reviewer #4. The unwinding of Helix G, which is centred upon Lys216, and a change in the conformation of Lys216, are exactly the same thing. The sentence Reviewer #4 objects to reads:

“Moreover, the unwinding of helix G in bR frees the carbonyl oxygen of Lys216 from its H-bond interaction to the amide nitrogen of Gly220 and provides a site for water a molecule (Wat453) to transiently bind.”

This sentence is accurate and has not been changed.

Presumably, literature values of the time constants are given in Fig. 1, the references should be added to the figure legend 1c.

The previous photocycle figure was a chimera of various publications. For the sake of providing a citation (which is a good suggestion by Reviewer #4), the image has been revised to be consistent with that drawn in Inoue et al., BBA 1837, 562 (2014). That article also cites primary sources for that their photocycle summarizes.

Page 6: “Whereas the high multiplicity of SSX data can sometime reveal unresolved electron density in the loops ...”. This is not correct. High multiplicity is necessary but not sufficient, it affects data quality. SSX data quality cannot be better than that of good rotation data. The “extra” electron density sometimes visible in SSX electron density maps originates from differences in structural distributions at ambient and cryogenic temperature or from the influence of cryoprotectants.

Reviewer #4 is mistaken to state that this sentence is not correct. This sentence is qualified by the phrase “..... can sometimes reveal unresolved electron density.....” and this is an experimental fact. All crystallographic data contain errors. In high-quality serial crystallography data, the power of averaging over data with a multiplicity of several hundred or several thousand reduces these errors. Nevertheless, we edited the offending sentence to read:

“Whereas high multiplicity serial crystallography data can sometimes reveal unresolved electron density in the loops of integral membrane proteins...”

Irrespectively, the point is moot since the second half of the sentence reads:

“this was not observed for NpSRII since the N-terminal end lacked electron density for the first 20 residues.”

Thus, this comment of Reviewer #4 has no bearing on the scientific findings of this work.

I obtain an average velocity of 780 $\mu\text{m/s}$ for the jet speed.

In the manuscript (page 8) it is stated that the jet has an average velocity of 760 $\mu\text{m/sec}$. This comes from: $(0.2 \mu\text{l/min}) / (60 \text{ sec/min}) / (\pi \times (0.075/2 \text{ mm})^2) = 754 \mu\text{m}$. Since we first divided this value by four to give a translation of 190 μm every 0.25 sec, this became rounded to 760 $\mu\text{m/sec}$ when 190 was multiplied by four. The value calculated by Reviewer #4 of 780 $\mu\text{m/sec}$ is very close to what we calculated, and certainly consistent within the accuracy with which such a value can be estimated.

Page 12 middle “density features associated with the Asp75 and Thr89 side-chains of NpSRII” -> should be Thr79

We thank Reviewer #4 for spotting this, and this correction has been made.

Page 16 Tyr204-> Thr204

We thank Reviewer #4 for spotting this, and this correction has been made.

PDB codes have changed and should be updated.

We thank Reviewer #4 for spotting this. All PDB codes are now updated.

In the dark structure, Wat 410 is cornered by positive and negative difference electron density, suggesting that it should move closer towards Asp75 and Lys205. Looking at the two structures and their densities accessible in the PDB there is still quite a bit of difference electron density in the active site.

As Reviewer #4 observes, there is some residual density associated with Wat410 in the released dark structure (pdb entry 7PNC), which can be corrected by a subtle shift of the position of this water molecule. This particular pdb file was first uploaded some time ago and has been updated as part of the revisions requested here.

Chloride in the dark structure is replaced by a water in the “steady state” structure, why?

Chloride was modelled in one of the original cryogenic temperature structures of SRII (Royant et al., PNAS 2001) but not the other (Luecke et al, Science 2001). This was always a matter of interpretation. However, since we follow Royant et al., for consistency we have updated all pdb files to include this chloride atom for the resting conformation but since R72 changes conformation, chloride is not modelled in the photoactivated state conformation.

Crystals. Description is inconsistent. Main text, p5 bottom: “... grew as needles with the longest dimension typically 100 μm or larger, and the smallest dimension less than 10 μm , but the longer needles tended to break during handling”.

Methods, p17: “These setups yielded crystals 40-60 μm in their largest dimensions, which were suitable for SSX”.

Longer crystals tended to break when collected and remixed with monoolein, which was done inside of a Hamilton syringe. This step is necessary for loading the sample into the sample reservoir prior to sample injection. Thus, there is no inconsistency, but rather a lack of accuracy in the description. This has been now clarified in the text as (page 5 and 6):

“Microcrystals grew as needles with the longest dimension typically up to 60 μm in length, and the smallest dimension less than 10 μm (**Extended data fig. 1**). However, the longer needles tended to break during handling as LCP microcrystal slurries were transferred into Hamilton syringes for adjusting the sample consistency prior to sample injection, and subsequently loading into the reservoir for sample delivery. Thus, prior to injection, microcrystals were typically up to 20 μm in their largest dimension.”

A micrograph showing the crystals should be added to the Supplement.

Photographs of these microcrystals have been added to Extended data figure 1.

Extended Data Fig. 2. Laser illumination should be 5 ms not 10 ms.

We thank Reviewer #4 for spotting this error in the schematic and this has been corrected.

It would be useful to add residue numbers to Fig. 4a, showing a direct connection between helices and amino acid regions.

This change has been made as requested. Since these figures are small, we did not label the water molecules, but they are labelled in the main text figures.

Reviewer #4

I appreciate the author's efforts to address my comments, in particular not only by writing but also by performing additional experiments. While most of my previous concerns have been addressed, some remain open and new ones have come up. Without addressing these I cannot support publication of the manuscript, unfortunately.

Major points

The authors claim that “unlike bR, helix G of NpSRII does not unwind near the conserved lysine residue...and therefore transient water molecule binding sites do not arise immediately to the cytoplasmatic side of retinal. The structural differences prolong the duration of the NpSRII photocycle relative to bR...”. I am still not convinced whether the structural differences between the two proteins in their photo-stationary state are indicative of functional differences, due to crystal constraints or a combination of both. The authors believe it is the former.

Our interpretation is based upon the difference electron density after illumination. For example, that shown in **Figure 6, a to c** is pasted below. The difference electron density observed for bR (panel a) and SRII (panels b and c) in this region of helix G (“near the conserved lysine”) are strikingly different. These differences between bR and SRII are unambiguous observations of our study.

To address my objection that it is difficult to correlate functional states observed in crystallo with kinetic features derived in solution the authors analyzed the photocycle kinetics with the protein dispersed in LCP (before crystallization) and after crystallization by difference spectroscopy. I realize the difficulty of such an experiment and laud the authors for doing so. The procedure is now well described and yields very good results for the non-crystallized protein in LCP (Fig. 2 a,b). Due to the poor signal of the crystalline sample the authors used the basis spectra derived from the non-crystallized protein in LCP for the analysis of the spectra of the crystallized protein in LCP. This is far from ideal but probably necessary. The same analysis was applied, using the same kinetic scheme, fitting two exponential decays. However, in the case of the crystalline sample I do not agree with the approach/result. It seems that the M state decays (yellow line in Fig 2) but I doubt that the O state (blue line) is formed. This is apparent from both derived numbers (essentially same decay times for the M/O crossover populations and the decay of the photoexcited population). Occam's razor should be applied and the data should be fitted with the minimal scheme, that is one

exponential decay. Then the residuals should be plotted. If they show an exponential like systematic decay, inclusion of the second exponential is warranted. If not, the second component is simply not there. If the latter is the case, the question arises whether the system can go through a functional photocycle in the crystal or whether the formation of the O state is simply so slow that it cannot be observed, at least within this observation window. More generally, with such kinetics, very slow formation of O followed by rapid decay, O will not accumulate and will be extremely hard (essentially impossible) to observe it with this experimental setup. Thus, my current conclusion of the analysis is that the data have been overfitted. This objection needs to be refuted. If I am not wrong, analysis needs to be repeated without any O state.

Figure 2f (pasted below) shows a positive feature between 520 nm and 650 nm in the difference spectrum from SRII microcrystals, and this is a fingerprint of an O-state. This cannot be explained by a pure M-state, since the mustard line in this figure (the M-state difference spectrum) cannot explain this positive feature. We therefore feel that Reviewer #4 is mistaken to suggest that we remove the O-state from the spectral decomposition of spectroscopic data recorded from microcrystalline slurries.

Fortunately, **Reviewer #5** writes: “As far as the time-resolved spectral data are concerned, I think the authors have adequately responded to the comments of reviewer 4 and the manuscript has been appropriately revised.” As we argued previously, spectroscopic measurements from microcrystalline slurries are not as reliable as for measurements from homogeneous samples.

The statement by the authors concerning my comment relating to crystal contacts is not correct (“Reviewer #4 is mistaken. As shown in Nango et al., Science 2016 (Figure S6), the EF loop and the C-terminus of a symmetry related molecule form crystal contacts in the bR P63 LCP crystals.”); it starts with errors in Nango et al. Citing from Nango et al, Science 2016: “This discrepancy may be reconciled by noting that motions of these helices are severely restricted in the P63 crystal form because residues 165 and 166 of the E-F loop participate in crystal contacts with residues 232 and 234 of the C terminus (fig. S7). “This statement is wrong, there are no crystal contacts between residues 165 and 166 of the EF loop and residues #232, 234. Residues 165/166 are close to a beta-turn from residues #70 to #74. It is unlikely that this closeness restricts the 165/166 movement, there is no H-bond or other strong interaction, just a van der Waals contact.

Concerning the C-terminus: Looking at 6rqp, the two C-terminal residues contact residues 129-131 of a symmetry-related (X-Y,X,Z+1/2) molecule, as well as residue 72 of the same symmetry-related molecule. In some of the other structures, such as 6g7h, the C-terminus is

one residue longer; that residues then also contacts 128-129 on that same symmetry mate, as well as residues 63-64, the latter coming from $-X, -Y, Z+1/2$. By contrast, as pointed out in my last review, in SR-II there are strong H-bonds between the backbone amide of Ser 154 and the backbone carbonyl of Gln#151 and between the backbone carbonyl of Glu151 and the backbone amide of Ser#154. This will definitely have an influence on mobility. Moreover, as pointed out in the last review, there is a strong H-bond between Tyr199 and the backbone carbonyl of #Ala125. This may affect the structural changes of the F-helix, including changes around Thr204. In bR light-induced changes (Weinert et al) extend all along the F-helix, including the region downstream of Val210 (corresponding to Tyr199 in SRII which is involved in the aforementioned crystal contact). In SRII the structural changes are generally smaller and strongly diminished in the helical turn around Ile197. Potentially the “anchoring” of Tyr199 could affect the structural changes of Tyr174/Asp201 (which are much larger in bR). Importantly, Tyr199 has been shown to form an essential H-bond with Asn74 (HtrII) in the complex between photosensor (SRII) and transducer (HtrII) (Gordelly, Nature 2002). Proton transport activity was shown to be blocked in SRI and latently in SRII by HtrI and HtrII binding, respectively (Schmies et al, 2001 PNAS, Sudo et al, Biophys J. 2001). Thus, “fixing” Tyr199 is worrisome.

In conclusion I do not agree with the argument of the authors (wrong as stated) that the two proteins face similar crystal contact restrictions. Whether this is the reason for the differences in structural changes/kinetics or the many amino acid substitutions (apart from the discussed Thr/Ala) is hard to say.

In twenty-five years of studies of light-induced structural changes in bacteriorhodopsin using LCP grown crystals (reviewed *e.g.* Wickstrand et al, *BBA* 2015), only one study (Weinert *et al*, *Science* 2019) modelled a large light-induced movement of helix F. All other studies had this motion suppressed because of crystal contacts. We later learned that additives used by Weinert *et al.* to make the LCP microjet flow more smoothly can lead to a change in the bR space group over time. Thus, there may have been a fortuitous softening of crystal-contacts in the study of Weinert *et al.* Irrespectively, the light-induced movements in bacteriorhodopsin near the middle of helix G that are shown in the illustration pasted above (**Figure 6, a to c**) are very reproducible (see *eg.* Wickstrand *et al*, *Ann. Rev. Biochem.* 2019; Nango *et al.*, *Science* 2016). Thus, both SRII and bR have crystal contacts that prevent a large movement of helix F, yet the observed difference electron density near helix G is very different in the two cases. Since it is not possible to perform a time-resolved X-ray diffraction study without crystals, scientific judgements based upon other factors are required. We argue that these observable differences are due to differences in sequence that have been selected by evolution because of function.

Diffraction data / Refinement

It should be indicated which dark data set belongs to which light-data set or what the difference is. Neither dark structure overlays perfectly with the dark structure in the illuminated crystal datasets, why is this?

During structural refinement, the model as a whole must first be placed in the unit-cell. Moreover, as implemented in Phenix, the coordinates of the dark conformation are restrained about the dark structure, rather than truly being held fixed. We have therefore edited the appropriate sentence in the manuscript to read: “Light illuminated structures were refined using partial occupancy refinement with an occupancy specified in **Table S2** and the complementary resting conformation restrained about the dark conformation.”

The bond length between Retinal-C15 and Nz-Lys205 is 1.38Å in the dark structure (9H1W) and 1.27 Å in 8PWP (0-30 ms) and the other time-resolved structures. What is the reason? Why was neither water nor retinal included in structure 9h1x?

At the request of Reviewer #4, we copied and pasted pdb files and communicated these to the editor, who forwarded these pdb files to Reviewer #4. In one case we copied the wrong pdb file, which was at a step prior to retinal and water molecules being inserted. We apologise for this, but the uploaded pdb files are correct.

Some extremely slight variations in bond-distances arose during refinement. We checked the pdb files that were uploaded to the pdb and the numbers in those files are consistent and do not match the numbers given by Reviewer #4. Thus, we are at a loss as to what problem is being described here. But we will check again that there are no errors in the uploaded pdb files.

Table S1

The number of unique reflections makes no sense. It appears as if the datasets in columns 3-8 are not merged since they contain roughly double the number of reflections as in column 1. But why is the number in column 2 so much lower than in column 1 despite the fact that the resolution is higher?

We thank Reviewer #4 for this careful reading of the table. There were errors in the table and we have gone through and corrected these. We apologize for this oversight.

It would be good to refine all structures to the same resolution, in particular the low resolution. Why was a 10 Å resolution cut chosen for the light-data? This should at least be explained in the methods section.

As mentioned above, there were errors in the table. This has now been corrected and the refined resolution domains are consistent.

The number of diffraction hits/indexing rate should be changed for the datasets in columns 3-8. Clearly the ~ 800,00 diffraction hits refer to all the data which were later binned into the various time-slices such as 0-30 ms, 30-60 ms, ... The number of diffraction hits per time slice should be indicated (ca 7,500) and the indexing rate should refer to this subset.

We removed the rows of diffraction hits and indexing rate since the important information is the number of indexed patterns. When these data were first analysed from the raw images, the settings gave a lot of false positives for hits and therefore the deleted values did not contain useful information.

Indexed image -> indexed lattice.

We changed this.

The difference in unit cell length is quite big between the dark and illuminated crystals (I assume this is the photo-stationary dataset). I only checked structure 9h1x. At least in this case the unit cell constants given in the header differ from the ones given in table S1 as does the Wilson B factor. Please double check all structures/table S1.

Reviewer #4 is correct. The unit cells are consistent for all structures there were (historical) errors in Table S1.

Table S2:

I understand that Table S2 lists the spectroscopically determined populations of the various species. However, “structural refinement occupancies” is misleading since it implies to me that this reflects the occupancies used for refinement. Clearly only two structures were refined/dataset, the dark population and an activated one, but not two activated ones. Please rewrite such that it becomes clear what is meant.

The title has been changed to: **Table S2: Population estimates from spectroscopic analysis.**

The values in the row: Corrected for translation, were used in structural refinement. This is foot-noted with †Correction to the population within crystals due to light-exposed sample moving out of the X-ray beam. These values were used to fix the crystallographic occupancy during structural refinement.

Minor points:

Authors. This very likely needs modifications. Takashi is the first name; in line with all the other authors it should be Takashi Tomizaki I think the affiliations of Tomizaki Takashi and Florian Dworkowski are wrong. For sure Tomizaki should be Paul Scherrer Institute / SLS or MX Team and not SwissFEL. Although Dworkowski has moved to SwissFEL recently it is highly likely that the experiment was performed when he was still a member of the MX group at the SLS.

After consultation with other authors, the addresses of those affiliated with the Paul Scherrer Institute have been changed and Takashi’s name has been changed.

Line 82: delete (i.e. a photostationary state) not appropriate here

We have deleted “(i.e. a photostationary state)” as requested.

Line 87: reveal how structural rearrangements (grammar issue, the sentence makes no sense)

We adjusted this sentence to read: “More specifically, our TR-SSX data reveal how the structural rearrangements in helix G near the retinal binding site are considerably smaller for *NpSR*II than those observed in bR.”

Line 104: handling as -> handling when

Change made as requested.

Line 106: loading into the INJECTOR reservoir

Change made as requested.

Line 121 Calpha atoms relative to -> from

“relative to” is correct so no change made.

Line 138: correlated with A photostationary state

Change made as requested.

Line 156 after they microcrystals -> delete they

The word “they” is changed to “the”.

Line 189-192: I would assume this is due to the low data quality/multiplicity. You could test this by refining a dark dataset consisting of the same number of lattices as the time-resolved light data.

The sentence reads: “Some heating induced disordering might be implied by the observation that the average B-factor is 54.0 \AA^2 for the dark data-set and 60.0 \AA^2 during continuous illumination (**Table S1**). However, the average B factor for the time-resolved data sets from 0 to 150 ms vary from 45.0 \AA^2 to 47.0 \AA^2 (**Table S1**). Since these values are lower than the dark data-set, sample to sample variation in crystals may also explain this observation.”

Looking at Table S1, there were many more diffraction images merged in the dark control and during continuous illumination than for the time-resolved data-sets. Thus, we think “sample to sample variation in crystals explains this observation” and reading more into it than this may be misleading.

Line 334-335“... This water molecule interacts with $N\epsilon$ of Trp171 (SRII) and $O\gamma$ of Thr204 in *Np*SRII, and Trp182 (bR) and $O\gamma$ of Ala215 in bR”. Fix the errors, the water molecule has the same kind of interactions in both proteins. It interacts with the backbone carbonyl (**not** $O\gamma$) of Thr204/Ala215.

Reviewer #4 is correct. This has been corrected in the manuscript.

Line 469: Table 1 -> Table S1

This change has been made.

Line 471: ... diffraction data (containing 835719 diffraction hits).

We see that Reviewer #4 is referring to the number of diffraction hits listed in Table S1. We decided to delete this row in the table, as well as the row for the indexing rate, since the hit finder seems to have found a lot of false-positives. It is the number of indexed images and multiplicity which matters.

Reviewer #5 (Remarks to the Author)

My specialty is time-resolved spectroscopy. Therefore, I read the manuscript of the paper more carefully, especially the part about time-resolved absorption spectra. As far as the time-resolved spectral data are concerned, I think the authors have adequately responded to the comments of reviewer 4 and the manuscript has been appropriately revised. I do not find any problems in the revised manuscript.

We thank Reviewer #5 for this clear statement. We appreciate that Reviewer #5 acknowledges our work to incorporate the constructive input of Reviewer #4.

Reviewer #6 (Remarks to the Author)

The authors report a time-resolved serial synchrotron X-ray crystallography (SSX) study of *Natronomonas pharaonis* rhodopsin II (NpSRII), observing phototriggered conformational changes on the timescale of 10s to 100s of milliseconds. The motivating biological question is to determine how related rhodopsins alter the structural and dynamical consequences of retinal photoisomerization to accomplish diverse downstream outcomes, from vision to ion pumping. The motivating technical goal is to determine how effectively SSX can be used to perform longer timescale time-resolved crystallography studies in order to broaden access to the technique by making it less dependent on X-ray free electron laser (XFEL) resources. In summary, I think this is an interesting and well-performed study.

We thank Reviewer #6 for this accurate summary of scientific and technical objectives of the work.

I note that I was not involved in the initial review of the manuscript and therefore I considered the revised document in light of the responses in the text and the authors' rebuttal. The most substantive of the several revisions appears to be the inclusion of spectroscopic data that supports the authors' structural conclusions. I note that these experiments are non-trivial to perform and analyze-the authors have clearly made a serious, good-faith effort to address prior critiques. I also agree with Reviewer 2 that these data add valuable information to the manuscript that allows a clear correlation to be made between crystallographically- and spectroscopically-observed intermediates. Reviewer 4 makes several comments about important technical issues that I feel that the authors have adequately addressed, particularly with regards to the correlation between kinetics of intermediates in crystallo vs in solution. It is well-established that there are differences in these rates. I concur with Reviewer 4 that any such discrepancy merits close consideration, however I believe that the authors have done all that they can to address this. Given the extensive prior review, my minor comments below are offered in the spirit of polishing edits/considerations for a manuscript that, in my opinion, reports important work of potentially broad interest to the time-resolved structural biology community.

We thank Reviewer #6 for this clear and accurate report on the work. R.N. particularly appreciates the statement: *"the authors have clearly made a serious, good-faith effort to address prior critiques."*

Minor points:

Line 156: "they" is a typo.

Corrected

Lines 187-192. The B-factor comparison is potentially informative, but I feel the more interesting observation is that the unit cell parameters for the continuously illuminated crystals are slightly larger than the others. This expansion is often observed during crystal heating (e.g. Wolff et al, PMID: 37723259) and may be the most relevant observation to make about potential heating. However, the other timepoints have identical cells, and perhaps these parameters were held constant during scaling. If so, that would weaken the argument that the cell of the continuously illuminated sample is meaningfully larger than others.

Serial crystallography data provides are distribution of unit cell parameters within a narrow distribution. These parameters can be either fixed or allowed to vary, and this is a choice which must be made during data-processing. Unfortunately, there were misleading errors in the table. In reality, we used consistent unit cell parameters throughout: $a = 89.75 \text{ \AA}$, $b = 131.7 \text{ \AA}$, $c = 51.0 \text{ \AA}$. This has been corrected.

Lines 250-251 and elsewhere. Positive and negative peaks are mentioned and it might be helpful to mention what the sigma and $e^{-}/\text{\AA}^3$ values are for these peaks, particularly when comparative comments are being made about stronger vs. weaker features.

Following this suggestion, we have edited the text to include the maximum and minimum difference electron density peak values on selected waters and the carbonyl oxygen of the conserved lysine of helix G.

Line 289: “move in unison” indicates a specific correlated motion that is compatible with the data but not conclusively established by it. Perhaps this should be qualified with “consistent with”... or similar.

The sentence in the manuscript is changed to: “Difference electron density features associated with the Asp75 and Thr79 side-chains of *Np*SRII are consistent with these residues moving in unison (Fig. 5b,c),”

Line 357: What is the magnitude of the coordinate error referred to here? Is it Luzzati error, Cruickshank DPI, or something else? This should be clarified.

We have edited this sentence to now read: “Structural refinement suggests that the Trp171-WatA1 H-bond is lost for $\Delta t = 60$ to 90 ms (Fig. 6f) but this may be due to the limitations of partial occupancy refinement when trying to place two overlapping water-molecules with complementary occupancy at 2.4 \AA resolution.”

Lines 474-475. The sentence “Datasets were truncated using CCP4 TRUNCATE” is potentially confusing. I acknowledge that the program is called TRUNCATE, but crystallographers do not usually talk about truncating data so much as scaling, converting from intensities to amplitudes, etc. It would be better to explicitly say what TRUNCATE was used for here.

This sentence has been rephrased to read: “Datasets were scaled and measured X-ray diffraction intensities were converted to structure factor amplitudes with appropriate errors using CCP4 TRUNCATE.^{59,60}”